# The Perceptual Bandwidth Bottleneck in Vision-Language Models: Active Visual Reasoning via Sequential Experimental Design

**Anjie Liu** [* 1]  **Ziqin Gong** [* 1]  **Yan Song** [2]  **Yuxiang Chen** [2]  **Xiaolong Liu** [3 4]  **Hengtong Lu** [5]  **Kaike Zhang** [5]
**Chen Wei** [5]  **Jun Wang** [2]

## Abstract

Visual perception in modern Vision-Language Models (VLMs) is constrained by a perceptual bandwidth bottleneck: a broad field of view preserves global context but sacrifices the fine-grained details required for complex reasoning. We argue that high-resolution visual reasoning is therefore not only semantic reasoning but also task-relevant evidence acquisition under limited perceptual bandwidth. Inspired by active vision and information foraging, we formalise this process as sequential Bayesian optimal experimental design (S-BOED), where an agent decides which visual evidence to acquire before answering. Since exact Bayesian inference is intractable in continuous gigapixel spaces, we derive a tractable coverage–resolution objective as a proxy for task-relevant information gain. We instantiate this framework with FOVEA, a training-free procedure that refines VLM crop proposals through evidence-oriented probing. Experiments on high-resolution benchmarks show consistent gains over direct and ReAct-style baselines, with particularly strong improvements in search-dominated remote-sensing settings.

## 1. Introduction

Vision-Language Models (VLMs) have significantly advanced general visual understanding, demonstrating a remarkable ability to reason about holistic scene context (Bai

et al., 2025; Comanici et al., 2025). However, a critical performance gap remains: despite their high-level reasoning capabilities, these models often exhibit "perceptual blindness" in tasks requiring fine-grained resolution (Campbell et al., 2024; Li et al., 2025b). Current state-of-the-art models frequently struggle with small-scale object counting, optical character recognition (OCR), and precise spatial localisation, failing even when the underlying logic of the task is straightforward (Zhang et al., 2024; Tong et al., 2024). We argue that such failures are not only failures of semantic reasoning but also failures of evidence acquisition under limited perceptual bandwidth.

**The Perceptual Bandwidth Bottleneck.** We identify this limitation as a *perceptual bandwidth bottleneck*. Most standard vision encoders, such as ViT-based models, project an input image into a fixed number of visual tokens regardless of its original resolution (Dosovitskiy, 2020; Liu et al., 2023). This fixed budget induces an unavoidable field-of-view–resolution trade-off: a global view preserves broad spatial context but compresses fine-grained details, while a local crop preserves details but sacrifices coverage. When processing a high-resolution scene globally, each token must aggregate a large spatial area, causing small objects, text, and local spatial relations to vanish before reasoning begins. Consequently, the model cannot reason about evidence that is absent from its visual representation.

**The Need for an Active Strategy.** Alleviating this bottleneck requires the model to act, not merely to perceive. Instead of passively encoding a single downsampled image, the agent must engage in information foraging (Pirolli & Card, 1999): it must decide where to allocate high-resolution visual bandwidth in order to acquire task-relevant evidence. Passive scanning strategies, such as sliding windows, are computationally prohibitive and introduce large amounts of distractor evidence. Recent latent Chain-of-Thought (Li et al., 2025a; Sun et al., 2025) and tool-based methods (Ma et al., 2025; Zhang et al., 2025b; Su et al., 2025a; Gao et al., 2025) show that visual agents can benefit from iterative perception, but their crop or tool-selection policies often remain heuristic. They lack a decision-theoretic objective for deciding which observation is most

*Equal contribution . Code available at https://github.com/iamlilAJ/active-vlm. [1]The Hong Kong University of Science and Technology (Guangzhou), Guangzhou, China [2]University College London, London, United Kingdom [3]ShanghaiTech University, Shanghai, China [4]AI Lab, The Yangtze River Delta, China [5]Li Auto, Beijing, China. Correspondence to: Yan Song <yan.song.24@ucl.ac.uk>.

*Proceedings of the 43rd International Conference on Machine Learning*, Seoul, South Korea. PMLR 306, 2026. Copyright 2026 by the author(s).

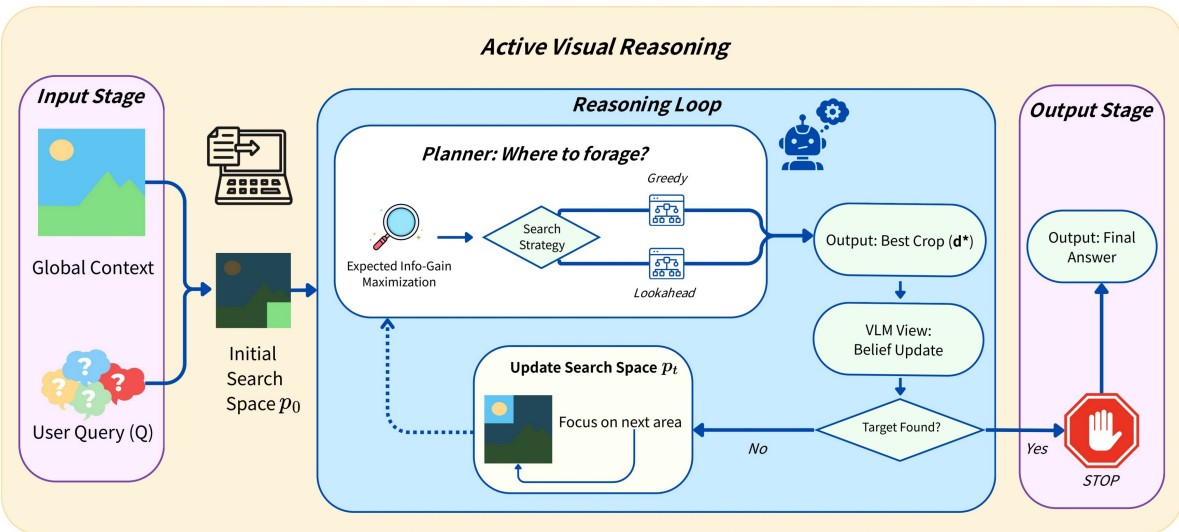

*Figure 1.* **S-BOED-guided active visual reasoning.** Under the perceptual bandwidth bottleneck, FOVEA iteratively refines VLM crop proposals to acquire task-relevant evidence. Candidate crops are scored by a coverage–resolution utility estimated through resolvability probing, and selected views update the interaction history for subsequent search.

valuable when the target is not immediately visible.

**Our Approach: Active Visual Reasoning as S-BOED.** We formalise active visual information acquisition as a sequential Bayesian optimal experimental design (S-BOED) problem (Lindley, 1956; Chaloner & Verdinelli, 1995; Rainforth et al., 2024). Analogous to a scientist choosing experiments to reduce uncertainty about hidden hypotheses, a VLM agent selects foveation actions to reduce uncertainty about the user's query, as illustrated in Figure 1.

This formulation exposes a key challenge overlooked in prior work: active visual reasoning is not just discrete visual tool selection, but continuous visual foraging under a bandwidth constraint. While BOED has recently been applied to discrete information-gathering tasks such as question selection (Kobalczyk et al., 2025; Choudhury et al., 2025), high-resolution visual reasoning requires selecting continuous foveation actions over large image spaces. The perceptual bandwidth bottleneck creates an *Information Cliff*: a wide view offers context but too little resolution, while a random zoom offers resolution but may miss the target. As a result, individual observations can have near-zero value until a critical coverage–resolution threshold is reached, motivating non-myopic planning.

Since exact Bayesian inference and exact expected information gain are intractable in continuous gigapixel spaces, we derive a tractable *Coverage–Resolution Objective* as a proxy for task-relevant information gain. We then instantiate the framework with *FOVEA*, a training-free inference-time procedure for Foveated Observation and Visual Evidence Acquisition. FOVEA treats the VLM's initial crop proposal as a noisy spatial prior, generates candidate foveations, probes

their query-relevant resolvability, and selects the design that maximises the coverage–resolution objective. Different optimisation strategies, including greedy sampling, MCMC-style refinement, and look-ahead planning, can be plugged into the same S-BOED-guided template.

The main contributions are: **(1) Problem formulation.** We identify the *perceptual bandwidth bottleneck* as a central obstacle in high-resolution VLM reasoning and formulate active visual reasoning as an S-BOED problem. **(2) Objective and instantiation.** We derive a tractable *Coverage–Resolution Objective* as a proxy for task-relevant information gain, and instantiate it with FOVEA, a training-free crop-refinement procedure. **(3) Empirical validation.** We show consistent gains over direct and ReAct-style baselines on high-resolution benchmarks, with further analysis of remote-sensing search, oracle gaps, proposal-limited failures, and compute–accuracy trade-offs.

## 2. Problem Formulation: Active Vision as Experimental Design

We ground our approach in the rigorous framework of Bayesian optimal experimental design (BOED). We consider a VLM agent performing active visual reasoning over a high-resolution image $I$ and a query $Q$. A comprehensive summary of notations is provided in Appendix B.

To bridge the gap between continuous visual signals and discrete token-based reasoning, we structure this formulation into three layers. First, we model the physical constraints of the VLM sensor, introducing the concept of perceptual bandwidth (Sec. 2.2). Second, we define the generative process,

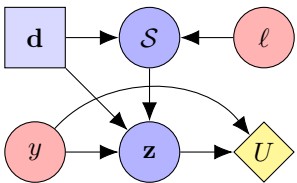

*Figure 2.* **Influence diagram of active visual reasoning.** The foveation design $\mathbf{d}$ and latent target location $\ell$ jointly determine the visibility event $\mathcal{S}$. This latent gate $\mathcal{S}$ modulates whether the observation $\mathbf{z}$ conveys information about the semantic target $y$. The agent's objective is to maximise the utility $U$, defined as the expected information gain over $y$, by actively managing the sensing design $\mathbf{d}$.

detailing how latent semantic states produce observable tokens through a resolution-gated mechanism (Sec. 2.3). Finally, we unify these components into a probabilistic graphical model (Fig. 2) that governs the agent's belief updates.

## 2.1. The Probabilistic System

Formally, we define the system as a tuple $\langle \boldsymbol{\theta}, \mathcal{D}, \mathcal{Z} \rangle$. Here, $\boldsymbol{\theta} \in \Theta$ represents the latent parameters (unknown world state), $\mathbf{d} \in \mathcal{D}$ denotes the design (action), and $\mathbf{z} \in \mathcal{Z}$ is the observation governed by a likelihood model $p(\mathbf{z} \mid \boldsymbol{\theta}, \mathbf{d})$.

## 2.2. Physical Constraints of Active Vision

To instantiate this framework, we first model the VLM as a stochastic sensor subject to rigorous resource limitations.

**Definition 2.1** (Perceptual Bandwidth $\mathcal{B}$). The fundamental bottleneck of VLM perception is the fixed encoder capacity (e.g., restricted token count (Dosovitskiy, 2020)), termed *perceptual bandwidth* $\mathcal{B}$. This capacity induces a *density-area trade-off* (Najemnik & Geisler, 2005), where the information density $\rho$ is defined as the ratio of the total bandwidth to the area $A(\mathbf{d})$ of a foveation crop:

$$\rho(\mathbf{d}) \triangleq \frac{\mathcal{B}}{A(\mathbf{d})}$$

**Definition 2.2** (Resolution Probability $\phi$). The probability that fine-grained features are resolved is governed by a saturation function $f_{\text{sat}}$ (e.g., sigmoid) of information density:

$$\phi(\mathbf{d}) \triangleq P(\text{Resolved} \mid \mathbf{d}) = f_{\text{sat}}(\rho(\mathbf{d})) \tag{1}$$

This creates a physical trade-off: larger crops (high $A$) suffer from low density ($\phi \to 0$), while smaller crops (low $A$) achieve high density ($\phi \to 1$).

*Remark* 2.3 (Analogy: The Semantic Nyquist Rate). The saturation behavior of $f_{\text{sat}}$ mirrors the classical Nyquist-Shannon Sampling Theorem (Shannon, 1949). We posit the existence of a critical density threshold $\tau_{\text{nyq}}$, termed the *semantic Nyquist Rate*. When $\rho(\mathbf{d}) < \tau_{\text{nyq}}$, the encoder fails to distinguish between distinct local features, rendering

fine-grained features indistinguishable. Conversely, once the density exceeds this threshold, the features become recoverable. In our framework, the sigmoid function serves as a differentiable approximation of this critical transition.

## 2.3. The Generative Process

Visual reasoning is not a static task but an interactive loop initiated by the agent's decisions. The generative process unfolds in three stages: action selection, physical interaction, and observation generation.

**Design Space: Foveation Actions** ($\mathcal{D}$). *Foveation actions* are parameterised as spatial crops $\mathbf{d} = [u, v, w, h] \in [0, 1]^4$. Crucially, $\mathbf{d}$ acts as a control variable for bandwidth allocation: by selecting a smaller region ($w \cdot h \ll 1$), the agent concentrates the fixed token budget onto a limited area, thereby boosting the local resolution density $\rho(\mathbf{d})$ and increasing the resolution probability $\phi(\mathbf{d})$.

**Latent Parameters: Semantic & Spatial State** ($\boldsymbol{\theta}$). We define the unknown state space as $\boldsymbol{\theta} \triangleq \{\ell, y\}$, which factorises into two components: the spatial location $\ell$ of the relevant object and the semantic target $y$ (e.g., the class label or text answer).

**Agent's Belief State.** At any time step $t$, the agent's knowledge about the latent parameters $\boldsymbol{\theta}$ is captured by the joint posterior $p_t(\ell, y)$. In real-world visual reasoning, spatial location $\ell$ and semantic identity $y$ are often coupled (e.g., context implies location). However, maintaining a full high-dimensional joint posterior is computationally intractable for real-time inference.

**Assumption 2.4** (Factorised Belief Approximation). To ensure tractability during the sequential design process, we adopt a *mean-field approximation* (Blei et al., 2017), assuming that the spatial search and semantic identification are momentarily decoupled during planning:

$$p_t(\ell, y) \approx p_t(\ell) \cdot p_t(y).$$

**Discussion.** This approximation reduces complexity from the joint product space $\mathcal{L} \times \mathcal{Y}$ to separate spatial and semantic factors, at the cost of ignoring higher-order spatial–semantic correlations. We use this factorisation only as a planning approximation, not as a claim that the true posterior is independent. The sequential feedback loop can partially mitigate this bias, since new observations reshape both the spatial and semantic beliefs through the context.

Under this assumption, we maintain two distinct belief maps: (1) A *spatial belief* $p_t(\ell)$ over the image coordinate space $\Omega$, representing the agent's uncertainty regarding the object's location. (2) A *semantic belief* $p_t(y)$, representing the uncertainty regarding the target's identity (e.g., class distribution), initialised by the linguistic priors in $Q$. This

separation allows the agent to explicitly reason about "where to look" (spatial uncertainty reduction) as a distinct objective from "what it is" (semantic identification), enabling the tractable EIG derivation in Section 3.

The core physical constraint is that semantic information is inaccessible unless the target is physically captured. This interaction is modelled by the *visibility event* $\mathcal{S}$, which acts as a latent bottleneck between the world state and the sensor.

**Definition 2.5** (The Visibility Event). To bridge physical actions and semantic observations, we define a binary latent indicator $\mathcal{S} \in \{0, 1\}$. This event represents whether the queried object is successfully captured by the encoder. Visibility occurs if and only if the object is both spatially encompassed and perceptually resolved:

$$P(\mathcal{S} = 1 \mid \ell, \mathbf{d}) \triangleq \underbrace{\mathbb{1}[\ell \in \mathbf{d}]}_{\text{spatial coverage}} \times \underbrace{\phi(\mathbf{d})}_{\text{perceptual resolution}} , \quad (2)$$

where $\mathbb{1}[\cdot]$ is the indicator function, $\ell$ is the latent spatial location, and $\phi(\mathbf{d})$ is the resolution probability (Eq. 1).

**Observation Generation (z).** Finally, the visibility state $\mathcal{S}$ gates the information flow to the VLM. The generative process concludes with the emission of the observation $\mathbf{z}$, which is a mixture of signal and noise modulated by $\mathcal{S}$.

**Definition 2.6** (Observation Model: Resolution-Modulated Likelihood). The visual observation $\mathbf{z}$ is governed by a mixture model conditioned on the latent state of $\mathcal{S}$. By the Law of Total Probability over the visibility event, the likelihood $p(\mathbf{z} \mid \boldsymbol{\theta}, \mathbf{d})$ is defined as:

$$p(\mathbf{z} \mid \boldsymbol{\theta}, \mathbf{d}) = \sum_{s \in \{0,1\}} p(\mathbf{z} \mid y, \mathcal{S} = s, \ell, \mathbf{d}) P(\mathcal{S} = s \mid \ell, \mathbf{d})$$
$$= \underbrace{P(\mathcal{S} = 1 \mid \ell, \mathbf{d})}_{\text{Gate Open}} \cdot p(\mathbf{z} \mid y, \mathbf{d})$$
$$+ \underbrace{(1 - P(\mathcal{S} = 1 \mid \ell, \mathbf{d}))}_{\text{Gate Closed}} \cdot p_0(\mathbf{z} \mid \mathbf{d}),$$
$$\text{(3)}$$

where $p(\mathbf{z} \mid y, \mathbf{d}) \triangleq p(\mathbf{z} \mid y, \mathcal{S} = 1, \ell, \mathbf{d})$ denotes the informative signal distribution when resolved, which we treat as conditionally independent of $\ell$ given $\mathcal{S} = 1$. And $p_0(\mathbf{z}) \triangleq p(\mathbf{z} \mid \mathcal{S} = 0)$ denotes the background noise distribution. Note that $p_0(\mathbf{z})$ is independent of $y$ ($y \perp \mathbf{z} \mid \mathcal{S} = 0$), representing the fact that an unresolved observation contains no semantic information about the target.

The complete generative process and the resulting decision-theoretic structure are summarised in the influence diagram in Figure 2, which serves as the basis for our sequential strategy derivation in Section 3.

## 3. Active Visual Reasoning as S-BOED

Building on the generative process established in Section 2, we now formulate the S-BOED for active information foraging through a three-stage derivation. We first define the theoretical sequential objective and identify the "Information Cliff" that renders standard greedy strategies insufficient (Sec. 3.1). To overcome computational intractability, we then derive a closed-form *Coverage-Resolution* utility under specific assumptions (Sec. 3.2). Finally, we present the idealised Bayesian belief update, which clarifies how positive and negative visual evidence should reshape the search distribution.

Throughout this section, all beliefs and information quantities at step $t$ are conditioned on the interaction history $\mathcal{H}_{t-1}$. For compactness, we write $p_t(\cdot)$ and $H_t(\cdot)$ for history-conditioned beliefs and entropies, and omit the subscript $t$ in mutual-information terms when the conditioning is clear.

### 3.1. The Sequential Objective

The agent's goal is to select a sequence of designs $\mathbf{d}_{1:T}$ to reduce uncertainty about the latent state $\boldsymbol{\theta} = \{\ell, y\}$. We quantify uncertainty using the Shannon entropy $H(\boldsymbol{\theta})$.

**Expected Information Gain (EIG).** For a single step, the utility of a design $\mathbf{d}$ is the expected reduction in entropy or, equivalently, the mutual information between the observation and the parameters:

$$\text{EIG}(\mathbf{d}) \triangleq \mathcal{I}(\mathbf{z}; \boldsymbol{\theta} \mid \mathbf{d}) = H(\boldsymbol{\theta}) - \mathbb{E}_{\mathbf{z} \sim p(\mathbf{z} \mid \mathbf{d})}[H(\boldsymbol{\theta} \mid \mathbf{z}, \mathbf{d})].$$

**Sequential Planning via Bellman Equation.** In the sequential setting, the agent maintains a history $\mathcal{H}_{t-1}$. The optimal strategy $\pi^*$ maximises the cumulative information gain over a horizon $T$. This is formally characterised by the *value function* $V^*$, which satisfies the Bellman equation:

$$V^*(\mathcal{H}_{t-1}) = \max_{\mathbf{d}_t} \Bigg( \underbrace{\text{EIG}(\mathbf{d}_t \mid \mathcal{H}_{t-1})}_{\text{immediate gain}}$$
$$+ \underbrace{\mathbb{E}_{\mathbf{z}_t \sim p(\mathbf{z} \mid \mathcal{H}_{t-1}, \mathbf{d}_t)}[V^*(\mathcal{H}_{t-1} \cup \{(\mathbf{d}_t, \mathbf{z}_t)\})]}_{\text{expected future value}} \Bigg).$$
$$\text{(4)}$$

Computing Eq. 4 requires solving a nested expectation over high-dimensional observations $\mathbf{z}$, which is computationally intractable. Furthermore, the structure of visual information poses a unique theoretical challenge:

*Remark* 3.1 (The Information Cliff). Standard active learning often assumes submodularity (diminishing returns) to justify greedy strategies. However, constrained active vision is often **super-additive**. Consider reading a small text: A wide view ($\mathbf{d}_{\text{wide}}$) locates the text but cannot read it ($\phi \to 0$); a random zoom ($\mathbf{d}_{\text{zoom}}$) can read but misses the

location ($\ell \notin \mathbf{d}$). Both yield zero gain individually. Only their sequence yields high information:

$$\mathcal{I}(y; \mathbf{z}_{\text{wide}}, \mathbf{z}_{\text{zoom}}) \gg \mathcal{I}(y; \mathbf{z}_{\text{wide}}) + \mathcal{I}(y; \mathbf{z}_{\text{zoom}}).$$

This "information cliff" requires look-ahead planning.

### 3.2. Derivation of the Tractable Coverage-Resolution Objective

While the ideal agent optimises the sequential Bellman equation, the nested expectations over high-dimensional observations $\mathbf{z}$ render it computationally intractable. In this section, we derive a closed-form approximation for the immediate task-relevant information gain that drives our practical crop-selection strategy.

**The Joint Information Objective.** The ultimate goal of the agent is to resolve the user's query $y$. However, due to the physical coupling between "seeing" and "understanding", the agent must jointly reason about the full latent state $\boldsymbol{\theta} = \{\ell, y\}$. Theoretically, the total information gain decomposes into spatial and semantic components:

$$\mathcal{I}(\mathbf{z}; \ell, y \mid \mathbf{d}) = \underbrace{\mathcal{I}(\mathbf{z}; \ell \mid \mathbf{d})}_{\text{Localization Gain}} + \underbrace{\mathcal{I}(\mathbf{z}; y \mid \ell, \mathbf{d})}_{\text{Semantic Gain}}.$$

In our active vision setting, resolving $y$ strictly necessitates localising $\ell$. Rather than optimising these terms separately, we focus on maximising the *marginal mutual information regarding the semantic target* $y$.

**Decomposition via the Visibility Event.** Directly computing $\mathcal{I}(\mathbf{z}; y \mid \mathbf{d})$ is intractable. To simplify, we introduce the auxiliary visibility variable $\mathcal{S}$. We first introduce a crucial assumption regarding the VLM's self-calibration:

**Assumption 3.2** (Calibrated Visibility)**.** We assume the observation $\mathbf{z}$ encodes sufficient statistics to determine the visibility state $\mathcal{S}$ (e.g., the model can distinguish between "blurry/empty" and "resolved" content). Mathematically, this implies $H(\mathcal{S} \mid \mathbf{z}, \mathbf{d}) \approx 0$, which allows us to approximate $\mathcal{I}(y; \mathbf{z} \mid \mathbf{d}) \approx \mathcal{I}(y; \mathbf{z}, \mathcal{S} \mid \mathbf{d})$. This assumption is empirically supported by recent findings that large-scale foundation models exhibit high calibration regarding their own predictive uncertainty (Kadavath et al., 2022).

By the chain rule,

$$\mathcal{I}_t(y; \mathbf{z}_t, \mathcal{S} \mid \mathbf{d}) = \mathcal{I}_t(y; \mathbf{z}_t \mid \mathbf{d}) + \mathcal{I}_t(y; \mathcal{S} \mid \mathbf{z}_t, \mathbf{d}).$$

Since

$$\mathcal{I}_t(y; \mathcal{S} \mid \mathbf{z}_t, \mathbf{d}) \leq H_t(\mathcal{S} \mid \mathbf{z}_t, \mathbf{d}) \approx 0,$$

Assumption 3.2 gives

$$\mathcal{I}_t(y; \mathbf{z}_t \mid \mathbf{d}) \approx \mathcal{I}_t(y; \mathbf{z}_t, \mathcal{S} \mid \mathbf{d}).$$

We then decompose the right-hand side as

$$\mathcal{I}_t(y; \mathbf{z}_t, \mathcal{S} \mid \mathbf{d}) = \underbrace{\mathcal{I}_t(y; \mathcal{S} \mid \mathbf{d})}_{\text{Term 1}} + \underbrace{\mathcal{I}_t(y; \mathbf{z}_t \mid \mathcal{S}, \mathbf{d})}_{\text{Term 2}}.$$

We analyse these two terms based on the conditional independence properties established in Section 2:

**Term 1.** As illustrated in Figure 2, the visibility event $\mathcal{S}$ is structurally determined by the spatial parameters $(\ell, \mathbf{d})$ and sensor physics. Under the Factorised Belief Assumption (Assumption 2.4), the semantic identity $y$ is independent of the spatial location $\ell$ during the planning phase ($\ell \perp y$). Consequently, since $\mathcal{S}$ is a function of $\ell$, it follows that $y \perp \mathcal{S} \mid \mathbf{d}$. Thus, $\mathcal{I}(y; \mathcal{S} \mid \mathbf{d}) = 0$.

**Term 2.** We expand the second term using the definition of conditional mutual information:

$$\begin{aligned} \mathcal{I}_t(y; \mathbf{z}_t \mid \mathcal{S}, \mathbf{d}) = {} & P_t(\mathcal{S} = 1 \mid \mathbf{d}) \cdot \mathcal{I}_t(y; \mathbf{z}_t \mid \mathcal{S} = 1, \mathbf{d}) \\ & + P_t(\mathcal{S} = 0 \mid \mathbf{d}) \cdot \underbrace{\mathcal{I}_t(y; \mathbf{z}_t \mid \mathcal{S} = 0, \mathbf{d})}_{=0}. \end{aligned}$$

The second part vanishes because an unresolved observation ($\mathcal{S} = 0$) yields only background noise independent of $y$ ($y \perp \mathbf{z} \mid \mathcal{S} = 0$).

Combining these results, we define the *Semantic Information Gain* objective:

$$\tilde{\mathcal{I}}_t(\mathbf{d}) \triangleq P_t(\mathcal{S} = 1 \mid \mathbf{d}) \cdot \mathcal{I}_t(y; \mathbf{z}_t \mid \mathcal{S} = 1, \mathbf{d}). \quad (5)$$

**The Perfect Perception Approximation.** Eq. 5 remains difficult to compute. To proceed, we rely on the strong semantic extraction capabilities of modern VLMs.

**Assumption 3.3** (Ideal Observer / Entropy Collapse)**.** For planning tractability, we model the VLM as an *ideal observer*. We assume that if the target is successfully foveated and resolved ($\mathcal{S} = 1$), the VLM extracts semantic information with high fidelity, causing the conditional entropy of $y$ to collapse to zero:

$$H(y \mid \mathbf{z}, \mathcal{S} = 1, \mathbf{d}) \approx 0.$$

This implies that the information gain from a successful foveation is approximately equal to the prior uncertainty:

$$\mathcal{I}(y; \mathbf{z} \mid \mathcal{S} = 1, \mathbf{d}) = H(y) - H(y \mid \mathbf{z}, \mathcal{S} = 1, \mathbf{d}) \approx H(y).$$

*Remark* 3.4. This assumption reduces the complex objective of semantic disambiguation to a geometric objective of visibility maximisation. It implies that the search strategy is responsible for acquiring high-fidelity evidence, while the interpretation of that evidence is delegated to the backbone VLM. While this ignores cases of hallucination on clear images, it is a necessary condition for tractable planning in open-ended spaces.

Under Assumption 3.3, the successful-foveation information term satisfies $\mathcal{I}_t(y; \mathbf{z}_t \mid \mathcal{S} = 1, \mathbf{d}) \approx H_t(y)$. Substituting this into Eq. 5 gives

$$\tilde{\mathcal{I}}_t(\mathbf{d}) \approx H_t(y) \, P_t(\mathcal{S} = 1 \mid \mathbf{d}). \qquad (6)$$

The remaining term is the probability that the latent target location is both covered by the crop and resolved under the fixed perceptual bandwidth. Marginalising over the current spatial belief $p_t(\ell)$ gives

$$P_t(\mathcal{S} = 1 \mid \mathbf{d}) = \left( \int_{\mathbf{x} \in \mathbf{d}} p_t(\mathbf{x}) \, d\mathbf{x} \right) \phi(\mathbf{d}) \triangleq \mathcal{J}_t(\mathbf{d}). \quad (7)$$

Thus, $\tilde{\mathcal{I}}_t(\mathbf{d}) \approx H_t(y)\mathcal{J}_t(\mathbf{d})$. Since $H_t(y)$ is independent of the current design $\mathbf{d}$, maximising the task-relevant information gain reduces to maximising the coverage–resolution objective $\mathcal{J}_t(\mathbf{d})$.

**Proposition 3.5** (Task-Relevant EIG Approximation)**.** *Under the Factorised Belief Approximation (Assump. 2.4), the Calibrated Visibility Assumption (Assump. 3.2), and the Ideal Observer Approximation (Assump. 3.3), the task-relevant EIG about the answer variable $y$ satisfies*

$$U_t(\mathbf{d}) \triangleq \mathcal{I}_t(y; \mathbf{z}_t \mid \mathbf{d}) \approx H_t(y) \, \mathcal{J}_t(\mathbf{d}),$$

*where $\mathcal{J}_t(\mathbf{d})$ is the coverage–resolution objective defined in Eq. 7. Since $H_t(y)$ is independent of the current design $\mathbf{d}$, maximising $U_t(\mathbf{d})$ reduces to maximising $\mathcal{J}_t(\mathbf{d})$.*

**The Coverage–Resolution Product.** The objective $\mathcal{J}_t(\mathbf{d})$ has a simple interpretation. Visibility requires the latent target location $\ell$ to be both spatially covered by the crop and perceptually resolved under the fixed visual-token budget:

$$\mathcal{J}_t(\mathbf{d}) = \underbrace{\left( \int_{\mathbf{x} \in \mathbf{d}} p_t(\mathbf{x}) \, d\mathbf{x} \right)}_{\text{Coverage}} \times \underbrace{\phi(\mathbf{d})}_{\text{Resolution}}. \qquad (8)$$

The greedy design is therefore selected as $\mathbf{d}_t^* = \arg\max_{\mathbf{d}} \mathcal{J}_t(\mathbf{d})$. This objective makes the coverage–resolution trade-off explicit: larger crops cover more posterior mass but reduce effective perceptual resolution, while smaller crops increase resolution but risk missing the target.

### 3.3. Formal Bayesian Belief Update

The coverage–resolution objective depends on the current spatial belief $p_t(\ell)$. In the idealised Bayesian model, this belief would be updated explicitly after each observation. Although our practical implementation approximates this update implicitly through the interaction history, the formal update clarifies how positive and negative visual evidence should reshape the search distribution.

Upon executing the optimal design $\mathbf{d}_t^*$ and receiving observation $\mathbf{z}_t$, the agent updates its spatial belief map $p_t(\ell)$ using Bayes' rule:

$$p_{t+1}(\ell) = \frac{p(\mathbf{z}_t \mid \ell, \mathbf{d}_t^*) \cdot p_t(\ell)}{\mathcal{Z}_t}, \qquad (9)$$

where $\mathcal{Z}_t$ is the normalisation constant.

The core of this update is the *spatial likelihood function* $p(\mathbf{z}_t \mid \ell, \mathbf{d}_t^*)$. To derive this from the joint observation model (Definition 2.6), we marginalise over the semantic target $y$. Relying on the Factorised Belief Assumption (Assumption 2.4), which treats $y$ and $\ell$ as independent during the inference step, the likelihood simplifies to:

$$p(\mathbf{z}_t \mid \ell, \mathbf{d}_t^*) \approx \mathbb{E}_{y \sim p_t(y)} \left[ p(\mathbf{z}_t \mid \ell, y, \mathbf{d}_t^*) \right].$$

Substituting the mixture model from Eq. 3 into this expectation, the likelihood bifurcates based on whether the latent location $\ell$ falls within the crop region $\mathbf{d}_t^*$:

$$p(\mathbf{z}_t \mid \ell, \mathbf{d}_t^*) = \begin{cases} \phi(\mathbf{d}_t^*)\mathbb{E}_{y \sim p_t(y)}[p(\mathbf{z}_t \mid y, \mathbf{d}_t^*)] \\ \quad + (1 - \phi(\mathbf{d}_t^*))p_0(\mathbf{z}_t \mid \mathbf{d}_t^*) & \text{if } \ell \in \mathbf{d}_t^*, \\ p_0(\mathbf{z}_t \mid \mathbf{d}_t^*) & \text{if } \ell \notin \mathbf{d}_t^*. \end{cases}$$

**Interpretation and Negative Evidence.** The term $\mathbb{E}_{y \sim p_t(y)}[p(\mathbf{z}_t \mid y, \mathbf{d}_t^*)]$ represents the marginal likelihood of the observation given that the target is resolved, averaged over the agent's current semantic belief. It quantifies how well the visual observation $\mathbf{z}_t$ supports the hypothesis that *any* valid target $y$ is present in the crop.

Crucially, this structure enables updates via *negative evidence*. Consider the scenario where the agent scans a candidate region with high effective perceptual resolution ($\phi(\mathbf{d}_t^*) \approx 1$) but receives an uninformative observation (i.e., $\mathbf{z}_t$ matches the background noise $p_0$). For locations inside the crop ($\ell \in \mathbf{d}_t^*$), the likelihood collapses to the signal probability, which is vanishingly small for noise inputs ($\mathbb{E}_y[p(\mathbf{z}_t \mid y)] \ll p_0(\mathbf{z}_t)$). Conversely, for unvisited locations ($\ell \notin \mathbf{d}_t^*$), the likelihood remains high at the baseline noise level $p_0(\mathbf{z}_t \mid \mathbf{d}_t^*)$, reflecting consistency with the "not seen" state. Through normalisation, this discrepancy suppresses the probability mass within the visited area ($\mathbf{d}_t^*$) and effectively "pushes" the belief mass to the unvisited regions, driving exploration.

## 4. Algorithmic Realisation

The S-BOED formulation specifies a decision-theoretic objective, but exact inference is intractable in gigapixel image spaces. In particular, Eq. 8 depends on a spatial belief $p_t(\ell)$ over a continuous domain, an unknown resolution function $\phi(\mathbf{d})$, and, for non-myopic planning, expectations over future observations. We therefore instantiate the framework

with *FOVEA*, a training-free procedure for Foveated Observation and Visual Evidence Acquisition. FOVEA should be understood as a practical surrogate instantiation of the S-BOED view rather than an exact solver with explicit posterior maps or exact EIG computation.

FOVEA uses the interaction history $\mathcal{H}_t$ as a history-conditioned search state, so later crop proposals can depend on both positive and negative evidence from earlier views. Appendix E provides empirical evidence for this history-based calibration.

Operationally, FOVEA has two main components: it estimates crop utility with a resolvability probe, and it optimises this utility with greedy sampling, MCMC-style refinement, or look-ahead planning.

### 4.1. Resolvability Probing

Zero-shot visual grounding in high-resolution regimes remains prone to spatial inaccuracies and hallucinations (Xiao et al., 2025; Su et al., 2025a). We therefore treat the VLM's initial crop proposal as a noisy spatial prior rather than ground truth. Around this proposal, FOVEA samples candidate foveations and scores each crop independently.

We introduce a binary resolvability signal $r \in \{0, 1\}$, where $r = 1$ denotes that the crop contains sufficient query-relevant visual evidence for the VLM to answer. This signal is not an exact estimator of information gain; rather, it is an empirical surrogate for crop utility under the S-BOED view. We define

$$\hat{\mathcal{J}}(\mathbf{d}) \triangleq P(r = 1 \mid I_{\mathbf{d}}, Q) \approx P(\text{VLM}(I_{\mathbf{d}}, Q) = \text{``Yes''}), \tag{10}$$

which estimates whether a candidate achieves a favourable coverage–resolution trade-off for the current query.

### 4.2. Optimisation Strategies

Given $\hat{\mathcal{J}}(\mathbf{d})$, FOVEA supports different optimisers. The greedy variant selects the candidate with the largest immediate resolvability score and is used as the efficient default. MCMC-style refinement improves local search by iteratively perturbing the crop proposal. For tasks with an information cliff, where the value of a view depends on what it enables next, we use a FOVEA-Lookahead that scores a candidate by the estimated resolvability of its simulated next state:

$$\mathbf{d}_t^* = \operatorname*{argmax}_{\mathbf{d} \in \mathcal{D}_{\text{cand}}} \hat{V}(\mathbf{d}, \mathcal{H}_{t-1}).$$

This keeps the objective fixed while allowing the optimiser to vary with the compute budget and task difficulty.

---

**Algorithm 1** FOVEA: S-BOED-Guided Local Perceptual Refinement

---

1: **Require:** Global image $I_{\text{global}}$, query $Q$
2: **Input:** Initial crop proposal $\mathbf{d}_{\text{seed}}$
3: Generate a candidate pool $\mathcal{D}_{\text{cand}}$ around $\mathbf{d}_{\text{seed}}$, including the seed crop and local perturbations
4: **for** each $\mathbf{d}_i \in \mathcal{D}_{\text{cand}}$ **do**
5:     Extract crop $I_{\mathbf{d}_i}$
6:     Estimate utility $\hat{\mathcal{J}}(\mathbf{d}_i) \leftarrow P(r = 1 \mid I_{\mathbf{d}_i}, Q)$
7: **end for**
8: **if** strategy is LOOKAHEAD **then**
9:     $\mathbf{d}_t^* \leftarrow \operatorname*{argmax}_{\mathbf{d} \in \mathcal{D}_{\text{cand}}} \hat{V}(\mathbf{d}, \mathcal{H}_{t-1})$
10: **else**
11:     $\mathbf{d}_t^* \leftarrow \operatorname*{argmax}_{\mathbf{d} \in \mathcal{D}_{\text{cand}}} \hat{\mathcal{J}}(\mathbf{d})$
12: **end if**
13: $\mathbf{z}_t \leftarrow \text{VLM}(I_{\mathbf{d}_t^*}, Q)$
14: $\mathcal{H}_t \leftarrow \mathcal{H}_{t-1} \cup \{(\mathbf{d}_t^*, \hat{\mathcal{J}}(\mathbf{d}_t^*), \mathbf{z}_t)\}$
15: **return** $\mathcal{H}_t$

---

## 5. Experiments: The Tool-Integrated Agent

We instantiate the S-BOED framework with FOVEA, a plug-in inference-time module that intercepts the VLM's `crop` commands and refines them before tool execution. This refinement is essential because external vision experts (e.g., `OCR`, `Detection`, and `Segmentation`) are equally subject to the *perceptual bandwidth* bottleneck. Without high-resolution inputs, these tools struggle to resolve dense or minute features in down-sampled global views (Akyon et al., 2022; Singh et al., 2019). Consequently, the cropping operation serves as a fundamental bridge to deliver high-fidelity signals to both the reasoning VLM and downstream tools. FOVEA optimises this critical interface by refining the crop's spatial parameters to maximise its informative utility.

**Benchmarks.** We assess FOVEA on four benchmarks: **HR-Bench** (Team, 2024), **MME-RealWorld-Lite** (Zhang et al., 2024), **V*Bench** (Wu & Xie, 2023), and **CV-Bench** (Tong et al., 2024). These datasets cover fine-grained recognition, small-object search, and 3D reasoning, all of which require high-fidelity local information. Together, these benchmarks test whether active crop refinement can alleviate the perceptual bandwidth bottleneck across recognition, search, and spatial reasoning tasks.

**Baselines.** We compare FOVEA against three groups of baselines. Proprietary models such as GPT-5 and Gemini establish the performance frontier for multi-modal reasoning, while Thyme (Zhang et al., 2025b) represents SOTA RL-based methods for integrated tool-use. Qwen3-VL-30B-A3B-Instruct serves as our controlled foundation model, processing only down-sampled global views without an active loop. The ReAct agent uses the same backbone and tool interface, but directly executes the VLM-proposed crop commands without S-BOED-guided refinement.

*Table 1.* **Main results on multimodal benchmarks.** FOVEA is compared with proprietary models, prior visual agents, and controlled direct/ReAct baselines under matched backbone settings where applicable. **Bold** indicates the best result within each comparison block.

| Method | Backbone | MME-RealW | CV-Bench | V* | HR-Bench (4K) | HR-Bench (8K) | Mean |
|---|---|---|---|---|---|---|---|
| *State-of-the-Art & Prior Agents* | | | | | | | |
| Thyme (Zhang et al., 2025b) | Qwen-2.5-VL-7B | 55.2% | 78.4% | **82.2%** | 77.0% | 72.0% | 73.0% |
| GPT-5 | Proprietary | 55.0% | 84.9% | 77.0% | 78.1% | 75.5% | 74.1% |
| Gemini 2.5 Flash | Proprietary | **58.5%** | **87.3%** | 80.1% | **83.4%** | **80.9%** | **78.0%** |
| *Controlled Comparison (30B Backbone)* | | | | | | | |
| Direct | Qwen3-VL-30B-A3B-Instruct | 48.2% | 81.2% | 81.2% | 80.0% | 75.9% | 73.3% |
| ReAct Agent | Qwen3-VL-30B-A3B-Instruct | 51.1% | 81.3% | 83.8% | 80.8% | 78.3% | 75.1% |
| RAP (Wang et al., 2025) | Qwen3-VL-30B-A3B-Instruct | 40.8% | 72.2% | **86.4%** | 79.6% | **80.6%** | 71.9% |
| **FOVEA (ours)** | Qwen3-VL-30B-A3B-Instruct | **54.6%** | **84.8%** | 85.3% | **84.5%** | 79.2% | **77.7%** |
| *Controlled Comparison (8B Backbone)* | | | | | | | |
| Direct | Qwen3-VL-8B-Instruct | 47.6% | 84.5% | 76.9% | 74.5% | 70.9% | 70.9% |
| ReAct Agent | Qwen3-VL-8B-Instruct | 48.1% | 83.9% | 78.8% | 77.7% | 73.8% | 72.5% |
| **FOVEA (ours)** | Qwen3-VL-8B-Instruct | **49.9%** | **84.7%** | **83.6%** | **80.9%** | **75.4%** | **74.9%** |

**Implementation Details.** Our primary results in Table 1 use the efficient greedy instantiation of FOVEA. To maintain computational tractability across the full benchmark, we prioritise inference-time efficiency over the more exhaustive look-ahead planning. For each initial crop proposal $\mathbf{d}_{\text{seed}}$, we generate two local perturbations, yielding a three-candidate pool $\{\mathbf{d}_{\text{seed}}, \mathbf{d}_{\text{small}}, \mathbf{d}_{\text{large}}\}$, and perform $K = 3$ stochastic probes per candidate to estimate $\hat{\mathcal{J}}(\mathbf{d})$. The final action $\mathbf{d}_t^*$ is selected greedily (Algorithm 1). More computationally intensive strategies (MCMC, look-ahead) are reserved for challenging subsets (Section 5.2).

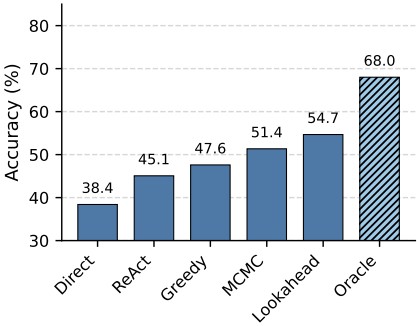

*Figure 3.* **Search efficacy in the gigapixel regime.** We compare Direct, ReAct, and FOVEA variants on the Remote Sensing subset against an oracle-crop baseline. FOVEA-Lookahead yields the largest gain, while the remaining oracle gap reflects residual backbone recognition and reasoning errors.

## 5.1. Main Results

Table 1 presents a comparative analysis across all evaluated benchmarks. On the 30B backbone, FOVEA achieves a mean score of 77.7%, improving over the ReAct baseline (75.1%) and the Qwen3-VL-30B-Instruct baseline (73.3%), and also surpassing the related active-perception baseline RAP (Wang et al., 2025) (71.9%). These results remain competitive with proprietary frontiers like Gemini 2.5 Flash (78.0%). To assess whether the framework transfers beyond

a single large backbone, we additionally evaluate it with the substantially smaller Qwen3-VL-8B-Instruct. The same overall trend holds: FOVEA improves the mean score from 70.9% / 72.5% (Direct / ReAct) to 74.9%, indicating that the proposed strategy is not specific to a single backbone scale. These gains indicate that better evidence acquisition can complement model scaling and heuristic tool use.

## 5.2. Strategy Efficacy in the Gigapixel Regime

To isolate the contribution of our search strategy from the VLM's semantic reasoning capabilities, we conduct a focused ablation on the **Remote Sensing** subset of MME-RealWorld-lite (Zhang et al., 2024). This setting is search-dominated: images are extremely large, targets are sparse, and task-relevant regions are often nearly invisible in the downsampled global view. We compare Direct, ReAct, and FOVEA variants against an oracle-crop baseline, where the VLM is given a human-annotated crop. The oracle does not represent perfect answering; rather, it separates *search failures* from *recognition failures*.

**Analysis.** As visualised in Figure 3, ReAct improves over the base model by enabling active tool use, but remains limited by noisy crop proposals. FOVEA-Greedy and FOVEA-MCMC further improve accuracy by refining local foveations, while FOVEA-Lookahead reaches 54.7%, compared with 45.1% for ReAct. The remaining gap to the oracle-crop baseline shows that evidence acquisition and backbone recognition are distinct bottlenecks: even with task-relevant crops, the VLM can still misrecognise or misreason. We analyse token costs in Appendix D.5, recognition failures in Appendix H.3, and qualitative cases in Appendix I.

## 5.3. Compute–Accuracy Scaling of Search Strategies

The previous section compares search strategies at a fixed budget on the full 150-question Remote Sensing set. To assess each strategy's *scaling potential*, we also vary the

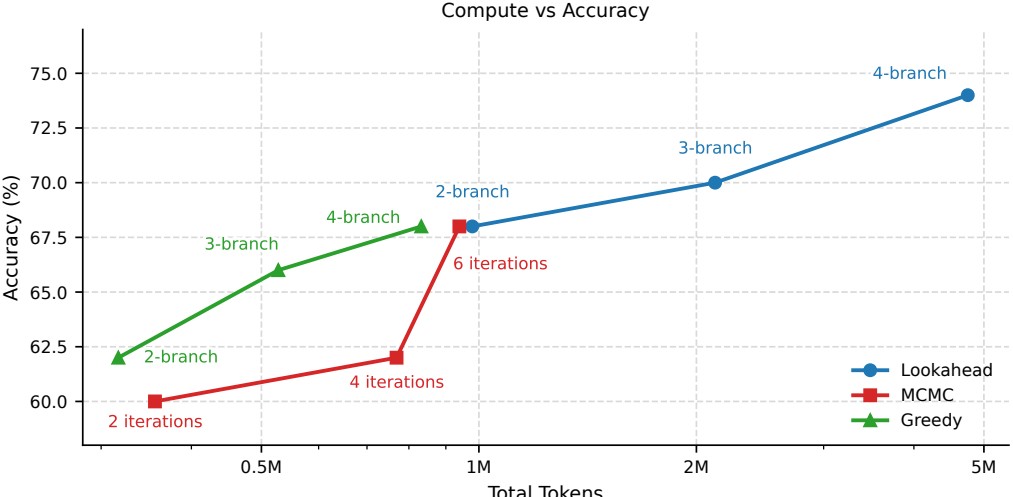

*Figure 4.* **Accuracy–compute scaling of FOVEA variants on a 50-example remote-sensing subset.** We vary the search budget within each policy family and report accuracy against average total tokens per question.

search budget within each policy family on a 50-example subset and plot accuracy versus average tokens per question (Figure 4). For FOVEA-Greedy, the budget is the number of sampled branches; for FOVEA-MCMC, the number of refinement iterations; and for FOVEA-Lookahead, the number of search branches.

**Analysis.** The trend is monotonic within each family: a higher search budget yields higher accuracy, but at increasing token cost. This indicates that FOVEA should be viewed as a family of compute–accuracy operating points rather than a single fixed policy. Lower-budget variants such as FOVEA-Greedy provide cheaper moderate gains and are suitable when latency is constrained, while higher-budget FOVEA-Lookahead yields larger improvements in search-dominated settings where additional search budget translates into meaningful gains.

This suggests that active perception provides a complementary axis of inference-time scaling: additional compute can be spent on acquiring higher-value visual evidence, not only on generating longer textual reasoning traces.

## 6. Conclusion

**Summary.** We formalise active visual reasoning under the perceptual bandwidth bottleneck as a sequential Bayesian optimal experimental design (S-BOED) problem. This perspective treats foveation not as heuristic preprocessing, but as principled acquisition of task-relevant visual evidence under a limited visual-token budget. We instantiate it with FOVEA, a training-free inference-time procedure that refines VLM crop proposals using a coverage–resolution utility estimated by resolvability probing. Experiments on high-resolution and gigapixel benchmarks show consistent gains over direct and ReAct-style baselines, especially in search-dominated remote-sensing settings. Overall, our results suggest that high-resolution VLM reasoning should be viewed as both semantic reasoning and active evidence acquisition.

**Limitations.** This work has three main limitations. First, the derivation relies on the *Ideal Observer* approximation (Assump. 3.3); in practice, even oracle-level crops may still cause backbone recognition or reasoning failures. Second, FOVEA is proposal-limited: if the relevant region never enters the candidate pool, local refinement and look-ahead cannot recover it. We analyse this cold-start bottleneck in Appendix H.2 and discuss multi-seed proposals as a mitigation. Third, resolvability probing and search add inference-time overhead, so FOVEA is best viewed as a compute–accuracy trade-off frontier rather than a fixed-cost policy.

**Future Work.** We identify three directions: (1) **Uncertainty Calibration.** Improving estimators of the VLM's epistemic uncertainty to sharpen task-relevant information-gain estimates (Choudhury et al., 2025; Feng et al., 2025). (2) **Amortised Inference.** Training a lightweight policy to predict useful foveation actions directly, using the coverage–resolution objective or downstream utility as supervision (Foster et al., 2019; Gershman & Goodman, 2014), thereby reducing the cost of iterative search. (3) **Adaptive Invocation.** The current implementation refines every crop call. A natural extension is a meta-policy that decides *when* active foveation is worth invoking, based on expected value of information (Howard, 1966) or causal necessity (Yu et al., 2026), connecting to the broader principle of *targeted intervention* (Liu et al., 2025; 2024a) that selective intervention is often more cost-effective than uniform guidance.

## Impact Statement

This work advances active perception and decision-making for multimodal agents. By formulating high-resolution visual reasoning as sequential experimental design, it provides a principled way for agents to decide what visual evidence to acquire before answering. This perspective may benefit applications where task-relevant information is sparse, local, or fine-grained, including remote sensing, document understanding, robotics, industrial inspection, medical imaging assistance, and embodied AI.

At the same time, more capable active perception systems may increase the reliability and scalability of automated visual monitoring and surveillance. Such systems should therefore be deployed with task-specific validation, transparency about acquired visual evidence, and appropriate human oversight in high-stakes settings. More broadly, our results suggest that improving multimodal agents requires not only stronger reasoning models, but also principled control over how models allocate perceptual resources.

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

# A. Related Work

**Active Vision.** The paradigm of controlling sensor parameters to maximise information utility originates from the seminal definition of *active perception* by Bajcsy (1988) and Aloimonos et al. (1988). Classical approaches focused primarily on geometric tasks, such as Next-Best-View (NBV) planning for 3D reconstruction (Connolly, 1985). In the era of foundation models, while the scope of active vision has expanded to semantic reasoning (Su et al., 2025b), modern VLMs largely remain passive observers, processing fixed-resolution images provided by humans (Dosovitskiy, 2020; Liu et al., 2023). We contend that for fine-grained reasoning, VLMs must reclaim the agency of active vision. We treat "zooming" or "cropping" not merely as image pre-processing, but as an *active sensory policy* analogous to biological foveation (Pirolli & Card, 1999), allowing the model to alleviate the effective-resolution limitations imposed by its fixed visual-token budget.

**Bayesian Experimental Design in Foundation Models.** Our framework is grounded in Bayesian optimal experimental design (BOED), a field established by Lindley (1956) to quantify the value of data via Expected Information Gain (EIG). While BOED has long been a staple in active learning for data selection (MacKay, 1992; Houlsby et al., 2011), its application to the inference process of foundation models is nascent. Recent works in NLP have begun to leverage BOED principles to address task ambiguity. Approaches such as Uncertainty of Thoughts (UoT) (Hu et al., 2024), Active Task Disambiguation (Kobalczyk et al., 2025), and BED-LLM (Choudhury et al., 2025) frame question-asking as an experimental design problem, shifting the cognitive load from *implicit heuristics* to *explicit inference* about the solution space. In this work, we extend this information-theoretic paradigm to the visual modality. We identify that for VLM agents, a primary source of epistemic uncertainty stems not from ambiguous prompts, but from the perceptual bandwidth bottleneck. Consequently, we adapt BOED to model the spatio-semantic search process, deriving a tractable strategy that optimises *foveation actions*—rather than textual questions—to resolve fine-grained visual ambiguities.

**Visual Agents and Inference-Time Scaling.** Early visual agents, such as VisProg (Gupta & Kembhavi, 2023) and LATTE (Ma et al., 2025), treated tool use as modular programming with static execution plans, lacking adaptability to dynamic perceptual uncertainty. More recent works focus on internalising policies via fine-tuning or Reinforcement Learning (RL), such as Thyme (Zhang et al., 2025b). While effective within their training distribution, these methods often function as black-box heuristics—they learn *where* to look via pattern matching, but lack an explicit model of *why* a region is informative. Our approach represents a shift towards a principled inference-time framework. By formalising active vision as an S-BOED problem, we decouple the decision-theoretic objective from the optimiser, using the coverage–resolution utility as a practical proxy for task-relevant information gain. This aligns with the emerging trend of inference-time scaling (Snell et al., 2024; Zhang et al., 2025a; Wang et al., 2024), and allows future research to plug in diverse optimisers—from greedy selection (Golovin & Krause, 2011) to MCMC sampling (Foster et al., 2019)—for active visual evidence acquisition.

## A.1. Comparison with Discrete BOED for LLMs

Our work is theoretically grounded in Bayesian optimal experimental design (BOED), similar to recent advances in LLMs such as Kobalczyk et al. (2025). However, applying BOED to visual reasoning introduces unique challenges that distinguish our S-BOED framework from text-based active disambiguation.

Kobalczyk et al. (2025) address *semantic ambiguity* in user prompts (e.g., "Write a code for X" where X is vague). Their agent selects discrete questions to partition the hypothesis space. In contrast, our work addresses *perceptual ambiguity* caused by sensor bandwidth. Our agent must select continuous spatial parameters to physically resolve the target. This shift from discrete question selection to continuous spatial foraging necessitates fundamentally different assumptions and objectives, summarised in Table 2.

*Table 2.* Comparison between discrete active task disambiguation and our S-BOED-guided visual evidence acquisition.

| Aspect | Kobalczyk et al. (2025) | FOVEA (Ours) |
|---|---|---|
| **Problem Domain** | **Semantic Disambiguation** Resolving vague user intent in text generation (e.g., code, 20 Questions). | **Perceptual Foraging** Resolving fine-grained visual features under bandwidth constraints. |
| **Action Space** | **Discrete & Enumerable** Selection from a finite set of generated candidate questions ($q \in \mathcal{Q}$). | **Continuous & High-Dimensional** Optimisation of spatial crop parameters in continuous coordinates ($\mathbf{d} \in [0,1]^4$). |
| **Belief State** | **Explicit Posterior** Estimated via explicit sampling of $N$ hypothetical solutions $\{h_i\}$. | **Implicit Context** Managed via the VLM's attention mechanism over the interaction history $\mathcal{H}_t$. |
| **Optimal Design** | **Space Partitioning** The optimal action bisects the solution space (e.g., a binary question with 50/50 split). | **Search & Resolution** The optimal action maximises the joint probability of coverage and resolution. |
| **Information Dynamics** | **Submodular (Diminishing Returns)** Every relevant question reduces entropy. Greedy strategies usually suffice. | **Super-additive (Information Cliff)** A crop that misses the target yields zero semantic info. Requires look-ahead planning. |
| **Constraint** | **Oracle Cost** Minimising the number of questions asked to the user. | **Perceptual Bandwidth** Minimising the loss of visual information due to token limits. |

# B. Notation

We align our notation with the standard Bayesian optimal experimental design (BOED) literature while introducing specific terms for the active visual reasoning setting. Table 3 summarises the primary symbols used throughout the paper. FOVEA-Greedy and FOVEA-MCMC are both myopic variants that optimise the immediate crop-utility estimator $\hat{\mathcal{J}}(\mathbf{d})$; they differ only in how candidate crops are generated. FOVEA-Lookahead is the non-myopic variant, using an estimated future value $\hat{V}$.

*Table 3.* Summary of notation.

| Symbol | Description | Domain / Definition |
|---|---|---|
| *Probabilistic Model & Generative Process* | | |
| $\boldsymbol{\theta}$ | Latent world state parameters | $\boldsymbol{\theta} = \{\ell, y\} \in \Theta$ |
| $\ell$ | Latent spatial location of the target | $\ell \in \Omega \subset \mathbb{R}^2$ |
| $y$ | Semantic target / answer variable | $y \in \mathcal{Y}$ |
| $\mathcal{S}$ | Visibility event / latent gating variable | $\mathcal{S} \in \{0, 1\}$ (Eq. 2) |
| $\mathbf{z}_t$ | Observation at step $t$ | $\mathbf{z}_t \in \mathcal{Z}$ |
| $p_t(\ell)$ | Spatial belief state at step $t$ | $p(\ell \mid \mathcal{H}_{t-1})$ |
| $p_t(y)$ | Semantic belief state at step $t$ | $p(y \mid \mathcal{H}_{t-1})$ |
| *Active Vision Setup* | | |
| $\mathcal{D}$ | Design space / foveation action space | $[0, 1]^4$ crop parameters |
| $\mathbf{d}_t$ | Selected foveation design at step $t$ | $\mathbf{d}_t = [u, v, w, h] \in \mathcal{D}$ |
| $I_{\mathbf{d}}$ | Crop induced by design $\mathbf{d}$ | Image region specified by $\mathbf{d}$ |
| $r_t$ | Resolvability signal | $r_t \in \{0, 1\}$ |
| $\mathcal{H}_t$ | Multimodal interaction history | $\{(I_{\mathbf{d}_\tau}, \mathbf{d}_\tau, \hat{\mathcal{J}}(\mathbf{d}_\tau), \mathbf{z}_\tau)\}_{\tau=1}^t$ |
| *Physical Constraints* | | |
| $\mathcal{B}$ | Perceptual bandwidth budget | Fixed visual-token / encoder capacity |
| $A(\mathbf{d})$ | Area of foveation crop | Normalised crop area |
| $\rho(\mathbf{d})$ | Effective perceptual density | $\rho(\mathbf{d}) = \mathcal{B}/A(\mathbf{d})$ |
| $\phi(\mathbf{d})$ | Resolution probability | $P(\text{Resolved} \mid \mathbf{d})$ |
| *Objectives & Optimisation* | | |
| $\text{EIG}_t^{\text{full}}(\mathbf{d})$ | Full Expected Information Gain | $\mathcal{I}_t(\mathbf{z}_t; \boldsymbol{\theta} \mid \mathbf{d})$ |
| $U_t(\mathbf{d})$ | Task-relevant information gain | $\mathcal{I}_t(y; \mathbf{z}_t \mid \mathbf{d})$ |
| $\mathcal{J}_t(\mathbf{d})$ | Coverage–Resolution objective | Theoretical proxy (Eq. 8) |
| $\hat{\mathcal{J}}(\mathbf{d})$ | Empirical crop-utility estimator | $P(r = 1 \mid I_{\mathbf{d}}, Q)$ (Eq. 10) |
| $V^*(\mathcal{H})$ | Optimal sequential value function | Bellman Eq. 4 |
| $\hat{V}(\mathbf{d}, \mathcal{H})$ | Estimated look-ahead value | Future resolvability estimate |
| *Practical Instantiation* | | |
| FOVEA-Greedy | Greedy local perturbation variant | Samples local candidates and selects $\arg\max_{\mathbf{d}} \hat{\mathcal{J}}(\mathbf{d})$ |
| FOVEA-MCMC | MCMC-style local search variant | Uses adaptive proposals to search for high-$\hat{\mathcal{J}}$ crops |
| FOVEA-Lookahead | One-step look-ahead variant | Selects crops by estimated future value $\hat{V}(\mathbf{d}, \mathcal{H})$ |

# C. Algorithms

---

**Algorithm 2** FOVEA-Greedy: Local Predictive Sampling

---

1: **Input:** Global view $I_{\text{global}}$, User query $Q$, Token budget $\mathcal{B}$
2: **Initialise:** $\mathbf{d}_{\text{prop}} \leftarrow \text{VLM}(I_{\text{global}}, Q)$ `// Initial region proposal`
3: **Perturbation:** Generate candidates $\mathcal{D}_{\text{cand}} = \{\mathbf{d}_{\text{prop}}, \mathbf{d}_{\text{small}}, \mathbf{d}_{\text{large}}\}$
$\quad$ *where* $\mathbf{d}_{\text{small}} = 0.8 \times \mathbf{d}_{\text{prop}}$ (high resolution), $\mathbf{d}_{\text{large}} = 1.5 \times \mathbf{d}_{\text{prop}}$ (high coverage).
4: **for** each $\mathbf{d}_i \in \mathcal{D}_{\text{cand}}$ **do**
5: $\quad$ Extract crop $I_{\mathbf{d}_i}$
6: $\quad$ $S_i \leftarrow P(r = 1 \mid I_{\mathbf{d}_i}, Q)$ `// Estimate crop utility`
7: **end for**
8: **Execution:** $\mathbf{d}^* = \arg\max_{\mathbf{d}_i} S_i$
9: **Return:** $\mathbf{d}^*$ to execute tool_call

---

---

**Algorithm 3** FOVEA-MCMC: Adaptive Crop Refinement

---

1: **Input:** Global view $I_{\text{global}}$, Query $Q$, Iterations $N$
2: **Utility Function:** $\hat{\mathcal{J}}(\mathbf{d}) = P(r = 1 \mid I_{\mathbf{d}}, Q)$
3: **Initialise:** $\mathbf{d}^{(0)} \leftarrow$ Random or VLM Proposal
4: **for** $t = 1$ to $N$ **do**
5: $\quad$ $\tilde{\mathbf{d}} \sim q(\cdot \mid \mathbf{d}^{(t-1)})$ `// Propose new region via Gaussian perturbation`
6: $\quad$ $\alpha = \min\left(1, \frac{\hat{\mathcal{J}}(\tilde{d}) + \epsilon}{\hat{\mathcal{J}}(d^{(t-1)}) + \epsilon}\right).$ `// Acceptance probability`
7: $\quad$ $u \sim \text{Uniform}(0, 1)$
8: $\quad$ **if** $u < \alpha$ **then**
9: $\quad\quad$ $\mathbf{d}^{(t)} \leftarrow \tilde{\mathbf{d}}$ `// Accept move to higher information density`
10: $\quad$ **else**
11: $\quad\quad$ $\mathbf{d}^{(t)} \leftarrow \mathbf{d}^{(t-1)}$ `// Reject move`
12: $\quad$ **end if**
13: **end for**
14: **Burn-in & Selection:** $\mathbf{d}^* = $ Mode or Best of $\{\mathbf{d}^{(t)}\}_{t=M}^{N}$
15: **Return:** $\mathbf{d}^*$

---

---

**Algorithm 4** FOVEA-Lookahead: One-Step Planning

---

1: **Input:** Global view $I_{\text{global}}$, Query $Q$, Proposal $\mathbf{d}_{\text{prop}}$, Scaling factors $\mathcal{S}$
2: **Initialise:** $\mathcal{D}_{\text{root}} \leftarrow \text{Perturb}(\mathbf{d}_{\text{prop}}, \mathcal{S})$
3: **for** each $\mathbf{d}_i \in \mathcal{D}_{\text{root}}$ **do**
4: $\quad$ `// Simulation (Expansion):`
5: $\quad$ $I_{\text{sim}} \leftarrow \text{Crop}(I_{\text{global}}, \mathbf{d}_i)$
6: $\quad$ $\mathbf{d}_{\text{next}} \leftarrow \pi_{\text{VLM}}(I_{\text{sim}}, Q)$ `// Predict agent's next move given` $\mathbf{d}_i$
7: $\quad$ `// Evaluation (Future Resolvability):`
8: $\quad$ **if** $\mathbf{d}_{\text{next}}$ is valid **then**
9: $\quad\quad$ $\mathcal{D}_{\text{leaf}} \leftarrow \text{Perturb}(\mathbf{d}_{\text{next}}, \mathcal{S})$
10: $\quad\quad$ $S_i \leftarrow \text{Mean}_{\mathbf{d}' \in \mathcal{D}_{\text{leaf}}} P(r = 1 \mid I_{\mathbf{d}'}, Q)$ `// Average utility of future leaves`
11: $\quad$ **else**
12: $\quad\quad$ $S_i \leftarrow 0$
13: $\quad$ **end if**
14: **end for**
15: **Execution:** $\mathbf{d}^* = \arg\max_{\mathbf{d}_i} S_i$
16: **Return:** $\mathbf{d}^*$

---

# D. Implementation Details

In this section, we detail FOVEA, the practical training-free instantiation of the S-BOED framework. We describe the agent architecture, the empirical crop-utility estimator, and the concrete search variants used in our experiments.

## D.1. Agent Architecture

Our system is built upon the ReAct agent framework. The core reasoning backbone is Qwen3-VL-30B-A3B-Instruct, a state-of-the-art multimodal model. The agent operates in a sequential loop: at each time step $t$, it receives the current visual observation and the interaction history. It then generates a structured "thought" trace followed by a specific tool invocation. The agent has access to four vision tools: cropping, detection, segmentation, and depth estimation. In the end, the agent aggregates both the ReAct trajectory and its direct reasoning to synthesise a final analysis and provide the answer. The system prompt is listed in Appendix J.

**Tool Specifications.**    We employ specific state-of-the-art backbones for the vision tools to ensure robust perception. For open-vocabulary object detection, we utilise Grounding DINO (Liu et al., 2024b); it accepts an image and a text prompt as inputs to output bounding box coordinates, and we draw these bounding boxes on the input image which will be returned to the agent with the coordinates. Segmentation is handled by SAM 2 (Ravi et al., 2024), which takes an image and prompt cues to generate high-quality pixel masks. We use different colours to mask the input image and return it to the agent. Depth estimation relies on Depth Anything (Yang et al., 2024), mapping a single RGB image to a relative depth map for spatial understanding. The agent will receive a heat map representing the depth along with the original image. Finally for OCR, we leverage MinerU (Niu et al., 2025), which processes an image region to extract and return structured text content.

**Tool Interception Mechanism.**    To implement active foraging without retraining the backbone model, we use a tool interception mechanism. The agent is provided with a standard `crop` tool definition, but when it invokes the tool with proposed coordinates $\mathbf{d}_{\text{prop}}$, the FOVEA module intercepts this call before execution. Instead of directly executing the raw proposal, FOVEA treats $\mathbf{d}_{\text{prop}}$ as a noisy spatial prior, refines it with one of the search strategies, and executes the refined crop $\mathbf{d}^*$ to return a more informative observation to the agent.

## D.2. Probabilistic Objective Realisation

To evaluate the coverage–resolution objective in a tractable manner, we employ the VLM itself as a stochastic evaluator of crop utility.

**The Empirical Estimator.**    We approximate the resolution probability $\phi(\mathbf{d})$ and the visibility event $\mathcal{S}$ by constructing a "resolvability verification" task. For a candidate design $\mathbf{d}$, we extract the corresponding image region $I_{\mathbf{d}}$ and prompt the VLM with the user query $Q$ alongside a specific system instruction (see Appendix J). The model is constrained to output a binary "Yes" or "No" indicating whether the region contains sufficient information to resolve the query.

**Monte Carlo Estimation.**    Since the VLM output is stochastic, we perform Monte Carlo estimation to smooth the objective surface. For every candidate design, we sample the model's response $N$ times. The estimator $\hat{\mathcal{J}}(\mathbf{d})$ is calculated as the expectation of the affirmative response. This score serves as an empirical crop-utility surrogate for the coverage–resolution objective, guiding the selection of the refined crop.

## D.3. Search Strategy Implementation

We implement three FOVEA variants corresponding to different search budgets.

**Greedy Perturbation Strategy.**    Algorithm 2 implements the computationally efficient FOVEA-Greedy variant. Instead of sampling from the entire image space, we generate a local search space $\mathcal{D}_{\text{local}}$ around the agent's initial proposal $\mathbf{d}_{\text{prop}}$ by applying a set of fixed scaling factors $\mathcal{C} = \{1.5, 1.0, 0.8\}$ to the proposed bounding box. The high-coverage variant (factor 1.5) expands the field of view to capture context that might have been narrowly missed by the initial proposal. The high-resolution variant (factor 0.8) zooms in further to maximise feature density, trading off spatial coverage. The candidate with the highest estimated score $\hat{\mathcal{J}}(\mathbf{d})$ is selected for execution.

**MCMC-based Adaptive Sampling.** Algorithm 3 implements the FOVEA-MCMC variant that treats the agent's initial proposal $\mathbf{d}_{\text{prop}}$ as the starting state $X_0$ of a Markov chain. We employ a Metropolis-Hastings framework with an adaptive step size to balance exploration and exploitation. For each iteration, we generate a candidate $\mathbf{d}'$ using a Gaussian perturbation. The step size is dynamically scaled based on the current crop's dimensions (setting $\sigma$ to 15% of the width/height), allowing the agent to perform coarse global shifts or fine local adjustments. We approximate the utility $U(\mathbf{d})$ using Eq. (10). A transition to $\mathbf{d}'$ is accepted if $\hat{\mathcal{J}}(\mathbf{d}') \geq \hat{\mathcal{J}}(\mathbf{d})$ or with a small probability (0.1) to prevent stagnation in local optima. We set a maximum of 6 iterations and utilise 3 stochastic probings per candidate. The process terminates early if a state achieves $\hat{\mathcal{J}}(\mathbf{d}) = 1.0$.

**Look-Ahead Planning Strategy.** Algorithm 4 implements the FOVEA-Lookahead variant, a one-step planner motivated by the Bellman update. For each candidate $\mathbf{d}_i$ in the local search space, the system simulates the crop and prompts the VLM to predict the subsequent action it would take given that observation. If the predicted next action is a further refinement (a sub-crop for example), we generate a hypothetical leaf node whose resolvability score will be evaluated. The value of the current candidate $\mathbf{d}_i$ is updated to reflect the expected gain of the future state, thereby favouring actions that lead to high-information states even if they do not immediately resolve the query.

### D.4. Hyperparameters

Table 4 lists the hyperparameters used across all experiments.

*Table 4.* Hyperparameters for FOVEA Inference

| Parameter | Value |
|---|---|
| Backbone Model | Qwen3-VL-30B-A3B-Instruct |
| Max Interaction Turns | 10 |
| Monte Carlo Samples | 3 |
| Perturbation Scaling Factors | $\{1.5, 1.0, 0.8\}$ |

### D.5. Inference Cost

Table 5 presents a cost-benefit analysis of the FOVEA variants. Active foraging introduces additional inference-time cost compared with the ReAct baseline, but the results reveal a clear compute–accuracy frontier. FOVEA-Greedy and FOVEA-MCMC incur moderate increases in input and output tokens due to candidate sampling and resolvability probing. FOVEA-Lookahead requires substantially more output tokens because it generates hypothetical future trajectories, but yields the largest accuracy gain. These results show that in search-dominated gigapixel regimes, additional compute for active perceptual planning can translate into meaningful accuracy improvements.

*Table 5.* Inference cost of search strategies compared to ReAct.

| Search Policy | Accuracy (%) | Accuracy Gain | Avg. Input Tokens / Query | Avg. Output Tokens / Query |
|---|---|---|---|---|
| ReAct (Baseline) | 45.1 | — | 46.5k (1×) | 0.3k (1×) |
| FOVEA-Greedy | 47.6 | 5.54% | 301.9k (6.5×) | 3.1k (9.8×) |
| FOVEA-MCMC | 51.4 | 13.97% | 359.3k (7.7×) | 4.0k (12.5×) |
| FOVEA-Lookahead | 54.7 | 21.29% | 441.4k (9.5×) | 17.5k (55.2×) |

## E. Empirical Evidence for History-Based Belief Calibration

In Section 4, we posit that the VLM can use interaction history to revise its search preference over candidate regions. This appendix provides empirical evidence for this history-based belief calibration, detailing the experimental setup, metric formulation, and quantitative results of a counterfactual intervention study.

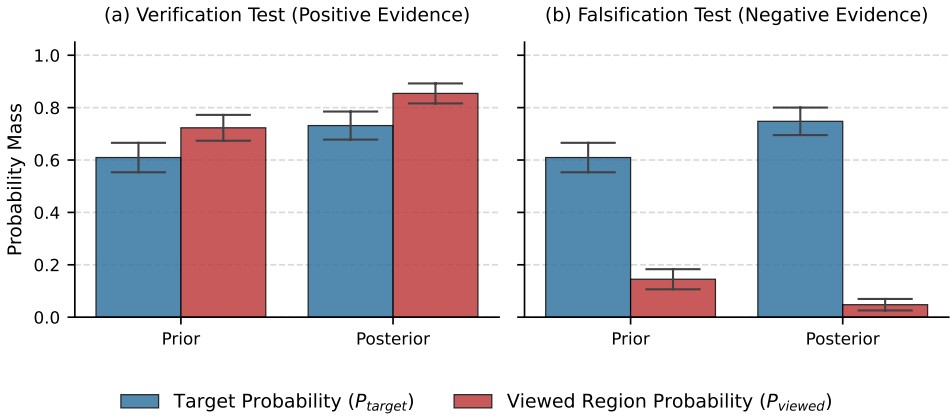

*Figure 5.* Bayesian belief update analysis across $N = 150$ samples. (a) Verification Test: Given positive evidence, the model concentrates probability mass on the target. (b) Falsification Test: Given negative evidence, the model prunes the viewed region (red bar collapses), shifting belief back to the valid search space. Error bars denote 95% CI.

### E.1. Experimental Setup

To rigorously test the model's ability to perform belief update, specifically *verification* (exploiting positive evidence) and *falsification* (pruning the search space given negative evidence), we utilised the 150 challenging visual search instances of the Remote Sensing subset of MME-RealWorld-lite. For each instance, we manually annotated:

1. **Ground Truth Grids** ($G_{GT}$): The set of $3 \times 3$ grid indices containing the target.

2. **Oracle Crop** ($C_{pos}$): A high-resolution crop perfectly centring the target object.

3. **Distractor Crop** ($C_{neg}$): A high-resolution crop of an irrelevant region.

We first queried the VLM with the global image to obtain a *prior* distribution over the nine grid regions. We then intervened by providing either the Oracle Crop (positive evidence) or the Distractor Crop (negative evidence) and queried the VLM for its *posterior* decision.

### E.2. Metric Formulation

To address quantisation artefacts where ground truth objects span grid boundaries, we aggregate probabilities rather than relying on single-label accuracy. We track two dynamic variables across the belief update process:

1. **Target Probability** ($P_{target}$): The total probability mass assigned to the true location of the object.

$$P_{target} = \sum_{i \in G_{GT}} P(\text{Grid}_i)$$

2. **Viewed Region Probability** ($P_{viewed}$): The total probability mass assigned to the region currently visible in the zoomed-in crop.

$$P_{viewed} = \sum_{j \in G_{overlap}} P(\text{Grid}_i)$$

where $G_{overlap}$ denotes the grid cells covered by the current intervention crop to a certain extent (15% of the grid area in our experiments).

### E.3. Quantitative Analysis

Figure 5 illustrates the aggregated results of the belief update experiment.

**Verification Test (Positive Evidence).** As shown in Figure 5 (a), providing the Oracle Crop triggers an obvious reduction in entropy. The VLM successfully identifies the evidence as sufficient, causing both $P_{target}$ and $P_{viewed}$ to converge towards 1. This confirms the model's capacity for *exploitation*, that is, when presented with the correct view, it confidently locks onto the target.

**Falsification Test (Negative Evidence).** Figure 5 (b) demonstrates the model's capacity for *exploration* and error correction. In the Prior state, the model assigns non-zero probability to the distractor region ($P_{viewed} > 0$). However, upon observing the detailed Distractor Crop, the probability assigned to this region collapses ($P_{viewed} \approx 0$). Crucially, the probability mass is not lost but is redistributed to the remaining search space, increasing the confidence in the true target $P_{target}$.

This behaviour—rejecting false evidence and reallocating probability mass to alternative regions—is consistent with the Bayesian-update view: the model can use positive and negative evidence in the interaction history to revise its search preference, rather than merely repeating its initial proposal.

## F. Probe Validity and Selector Ablation

A core component of our framework is the empirical utility estimator $\hat{\mathcal{J}}(\mathbf{d}) = P(r = 1 \mid I_{\mathbf{d}}, Q)$ defined in Eq. 10, which uses the VLM's Yes/No response on a candidate crop as a surrogate for crop utility. A natural concern, raised during review, is whether this signal genuinely tracks crop usefulness for the downstream task or merely reflects generic answerability confidence. This appendix presents two controlled diagnostics that address this concern. Both studies use the same 50 annotated remote-sensing position-reasoning examples from MME-RealWorld-Lite (Zhang et al., 2024).

### F.1. Probe Validity: Oracle vs. Distractor vs. Random Crops

**Setup.** For each example we compare three crop types: an *oracle* crop drawn from the human annotation that contains the answer-relevant region; a *distractor* crop drawn from a visually plausible but task-irrelevant region; and a *random* crop sampled uniformly from the image. For each crop we record (i) the Yes/No probe score $\hat{\mathcal{J}}(\mathbf{d})$, (ii) whether the crop centre falls inside the annotated oracle region (`hit_centre`), and (iii) the downstream QA accuracy when the model answers using the crop together with its normalised crop coordinates.

**Results.** Table 6 summarises the results. Oracle crops receive substantially higher probe scores than either distractor or random crops (0.633 vs. 0.187 / 0.187), and they also yield much higher downstream QA accuracy (52.0% vs. 10.0% / 12.0%). The effect size between oracle and distractor crops is large (Cohen's $d = 1.22$), indicating strong separation rather than a marginal trend. The probe score is also positively rank-correlated with both spatial grounding (Spearman $\rho = 0.538$ with `hit_centre`) and downstream correctness ($\rho = 0.392$ with QA accuracy).

*Table 6.* **Probe-score validity on annotated remote-sensing position-reasoning examples.** Oracle crops receive substantially higher Yes/No probe scores than distractor or random crops, and they also yield much higher answer accuracy when the model answers using the crop together with its normalised crop coordinates. The large effect size (Cohen's $d = 1.22$) indicates strong separation between oracle and distractor crops. The positive rank correlations ($\rho = 0.538$ with `hit_centre` and $\rho = 0.392$ with QA accuracy) indicate that higher proxy scores are associated with both better spatial grounding and better downstream task performance. `hit_centre` denotes that the centre of the candidate crop falls inside the human-annotated oracle region.

| Metric | Value |
|---|---|
| Oracle proxy score | $0.633 \pm 0.427$ |
| Distractor proxy score | $0.187 \pm 0.310$ |
| Random proxy score | $0.187 \pm 0.351$ |
| Oracle QA accuracy | 52.0% |
| Distractor QA accuracy | 10.0% |
| Random QA accuracy | 12.0% |
| Cohen's $d$ (oracle vs. distractor) | 1.22 |
| $\rho$(score, `hit_centre`) | 0.538 |
| $\rho$(score, QA accuracy) | 0.392 |

**Interpretation.** These results indicate that the Yes/No probe is not behaving as a generic "can the model produce an answer" confidence signal: if it were, oracle and distractor crops would receive similar scores, since the model can fluently

produce *some* answer in either case. Instead, the probe assigns markedly higher scores to crops that actually contain the answer-relevant evidence, and these higher scores translate into better downstream QA accuracy. We therefore use the probe as an empirical surrogate for crop utility, while explicitly noting that it is not an exact estimator of information gain.

### F.2. Selector Ablation: Probe vs. VLM-Direct vs. Random

**Setup.** The previous study evaluates probe scores on individual crops in isolation. To test whether the probe is also a better *selector* than alternative strategies under a fixed candidate pool, we construct a controlled pool by partitioning each image into a $3 \times 3$ grid of equal cells and compare three ways of selecting one cell: (1) **Probe**, which scores each cell independently with $\hat{\mathcal{J}}(\mathbf{d})$ and picks the highest-scoring cell; (2) **VLM-direct**, which presents all nine cells jointly to the VLM and asks it to pick the single best one in a single forward pass; (3) **Random**, which selects a cell uniformly. All three selectors operate on the *same* fixed pool, isolating the effect of the selection mechanism from candidate generation.

We evaluate three metrics on the selected cell: downstream QA accuracy, `hit_centre` (whether the selected cell's centre falls in the oracle region), and IoU between the selected cell and the oracle region.

**Results.** Table 7 reports the outcome. Probe-based selection outperforms both alternatives on all three metrics. In particular, Probe more than doubles the QA accuracy of VLM-direct selection (30% vs. 20%) and substantially improves localisation (`hit_centre` 52% vs. 34%; IoU 0.260 vs. 0.188).

*Table 7.* **Selector ablation on a fixed $3 \times 3$ candidate pool (50 remote-sensing position-reasoning examples).** For each image, we partition the full image into 9 equal grid cells and compare three ways of selecting one candidate cell: (1) **Probe**, which scores each cell independently with the Yes/No crop-utility question; (2) **VLM-direct**, which asks the model to choose the single best region among all 9 cells jointly; and (3) **Random**. Probe outperforms both alternatives in downstream QA accuracy, while also improving localisation quality (`hit_centre`) and average overlap (IoU).

| Selector | QA Acc. (%) | Hit Centre (%) | IoU |
|---|---|---|---|
| Random | 10 | 12 | 0.061 |
| VLM-direct | 20 | 34 | 0.188 |
| Probe | **30** | **52** | **0.260** |

**Interpretation.** Two observations are worth noting. First, the gap between Probe and VLM-direct shows that the gain does not come from candidate generation (the pool is identical) but from *how* candidates are scored: independently probing each crop at high resolution recovers more useful local evidence than asking the VLM to compare nine downsampled cells in a single forward pass. This is consistent with the perceptual-bandwidth view in Section 2.2, since the joint comparison forces the encoder to compress all nine regions simultaneously. Second, Probe improves not only QA accuracy but also `hit_centre` and IoU, indicating that the gain is grounded in better region selection rather than only in answer formatting.

## G. The Role of Intermediate Reasoning in Multi-Step Interaction

A natural concern with multi-step interaction is that the agent may accumulate *language drift*: by repeatedly generating intermediate conclusions and plans for the next view, the model could gradually commit to an incorrect hypothesis and reinforce it over subsequent steps. This appendix provides a simple ablation and a qualitative case showing that, on this class of questions, intermediate reasoning instead functions predominantly as an integration mechanism. Both studies use the same 50 remote-sensing position-reasoning examples introduced in Appendix F; absolute numbers in this appendix are therefore not directly comparable to the main-text Remote-Sensing results in Section 5.2, which use a different protocol on a different subset.

### G.1. Ablation: Evidence-Only Multi-Step Interaction

We compare the standard multi-step setting against an *evidence-only* variant in which all intermediate textual reasoning is removed from context, while the sequence of intermediate crops is kept identical. The two variants therefore receive the same visual information; the only difference is whether the model's intermediate reasoning trace is preserved.

As shown in Table 8, removing intermediate reasoning lowers accuracy from 66.0% to 48.0%. Since the visual evidence is unchanged, this 18-point gap is attributable to the role of intermediate reasoning in integrating that evidence across views.

*Table 8.* **Effect of removing intermediate textual reasoning in multi-step interaction.** Both variants share the same sequence of intermediate crops on the same 50 remote-sensing questions.

| Setting | QA Acc. (%) | Description |
|---|---|---|
| Full multi-step reasoning | 66.0 | Intermediate reasoning and crops |
| Evidence only | 48.0 | Intermediate crops only |

## G.2. Qualitative Case

Figure 6 shows a representative example. Without intermediate exploration reasoning, the model commits to the wrong quadrant in a single shot (answer D). With sequential scanning and local verification, the same backbone recovers the correct quadrant (answer C), illustrating that the multi-step trace can revise an early hypothesis rather than lock it in.

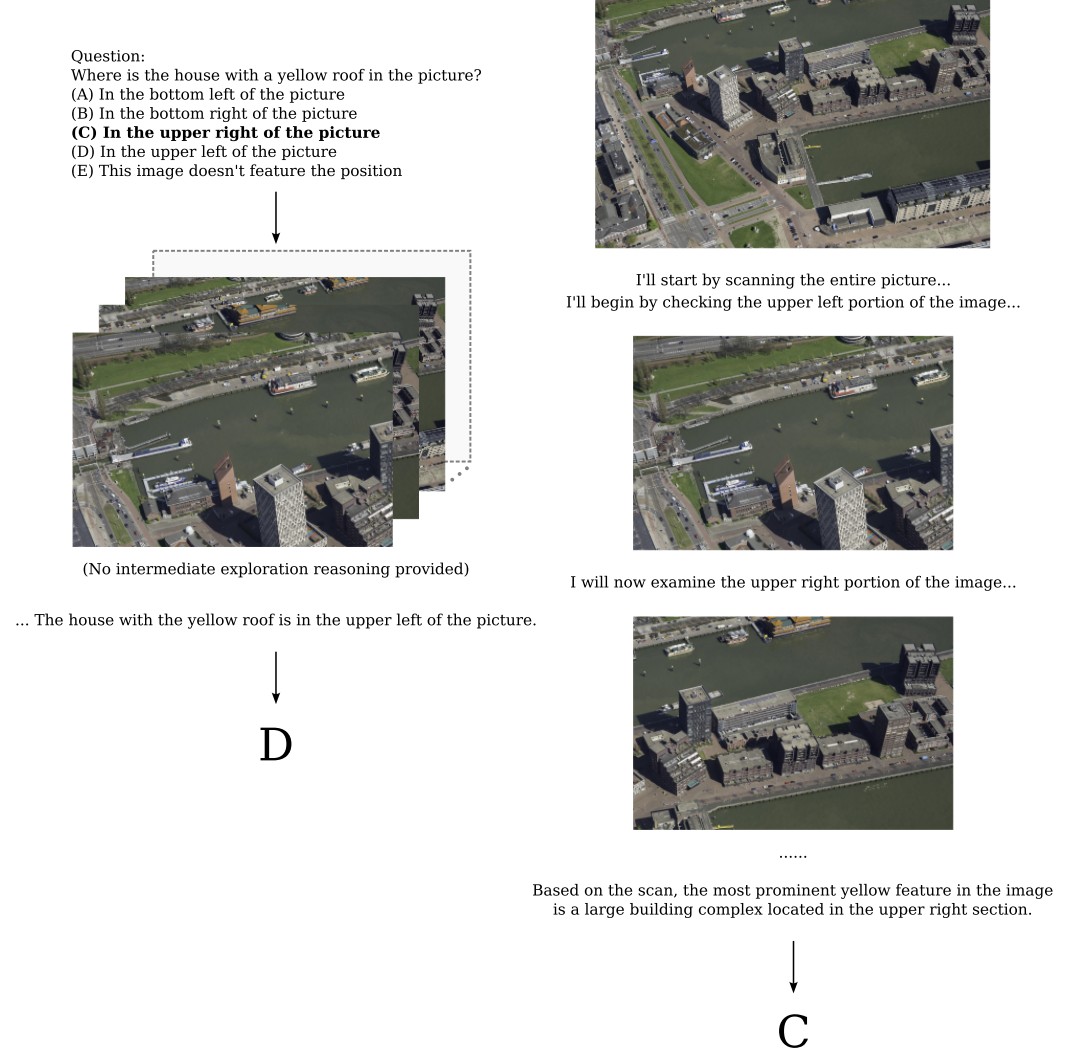

*Figure 6.* **Sequential reasoning revising an early hypothesis.** Without intermediate reasoning (left), the model predicts the wrong quadrant (D). With sequential scanning and local verification (right), it recovers the correct answer (C).

## H. Failure Analysis

As shown in Figure 3, while the FOVEA-Lookahead outperforms greedy baselines by simulating future belief states, a performance gap relative to the Oracle persists (54.7% vs. 68.0%). Since the oracle-crop baseline uses human-annotated crops that largely remove the search bottleneck, this gap isolates failures in *search dynamics* rather than recognition

capability. The remaining gap to 100% within the Oracle itself is attributable to a separate bottleneck, namely the backbone's reasoning reliability, which we discuss in Appendix H.3.

This appendix examines the search-side failures qualitatively (Section H.1) and then quantifies the dominant mode through a controlled multi-seed diagnostic (Section H.2).

## H.1. Failure Modes

Qualitative analysis of the Look-ahead error cases reveals two primary failure modes where the active search strategy diverges from the optimal path.

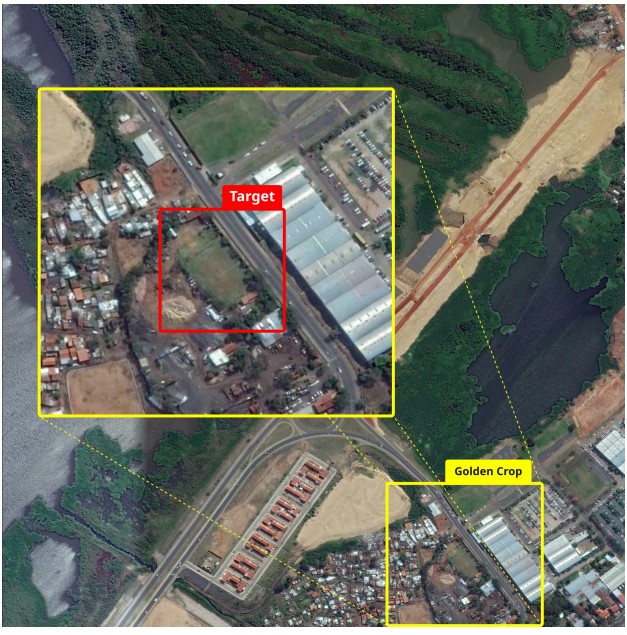 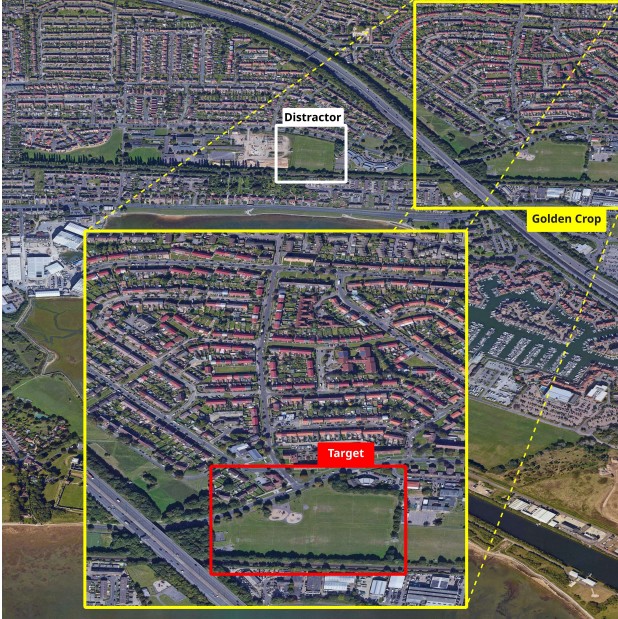

*(a)* Prior Misalignment (Cold Start)    *(b)* Semantic Distractor Traps

*Figure 7.* **Qualitative failure analysis.** (a) In the *Cold Start* scenario, the target (a white circular pattern) is imperceptible at the global scale, leading to a near-zero initial probability mass in the correct region. (b) In the *Distractor* scenario, the task requires finding the "largest football field". The agent exhausts its inference budget investigating visually similar fields (white box) before it can identify the true target (red box).

**Cold Start (Prior Misalignment).**   The efficacy of the Look-ahead search is bounded by the quality of the initial belief distribution. As shown in Figure 7a, small targets such as the white circular pattern of the target green rectangular grassland are often completely imperceptible in the downsampled global view. Consequently, the VLM assigns a negligible probability mass to the target region during the initial scan. The FOVEA-Lookahead, relying on entropy to guide exploration, interprets these false-negative regions as "resolved background" and directs its budget towards more ambiguous but incorrect regions. Unlike the Oracle, which is initialised with ground truth coordinates, the planner cannot recover from this initial misalignment.

**Semantic Distractor Traps.**   In dense remote-sensing imagery, the planner occasionally falls victim to semantic distractors: objects that share visual features with the target but do not satisfy the specific query constraints. Figure 7b illustrates this for the query "Find the largest football field". The environment contains multiple candidate fields; the distractor (white box) is visually salient and generates high expected information gain. The agent, limited by a finite inference budget, expends its steps zooming into and verifying these smaller fields. By the time the ambiguity is resolved (determining the field is too small), the episode terminates before the agent can explore the true target location (red box).

These cases highlight that while Look-ahead search effectively handles local spatial reasoning, it remains susceptible to global priors and budget constraints that the Oracle does not face.

## H.2. Quantifying the Proposal Bottleneck

The cold-start failure mode in Section H.1 is fundamentally a *proposal* bottleneck: when the global view does not surface the target, the single seed proposal $\mathbf{d}_{\text{seed}}$ is misaligned, and local refinement around it cannot recover a region that never enters the candidate set. To quantify how much of the Oracle gap is attributable to this bottleneck, we run a controlled diagnostic on 50 remote-sensing position-reasoning examples comparing two initialisation regimes under the same downstream scoring rule:

- **Single-seed** (current default): one VLM proposal, refined into 3 local candidates.

- **Multi-seed**: 9 seed proposals, each refined into 3 local candidates, yielding a candidate pool of 27.

We decompose each failure into one of three categories. A failure is *proposal-limited* if the correct region never enters the candidate set; *search-limited* if the correct region is present in the candidate set but is not selected; and *reasoning-limited* if a target-relevant crop is selected but the backbone still answers incorrectly. This decomposition isolates whether errors are caused by the candidate generator, the selector, or the backbone.

*Table 9.* **Failure decomposition under single-seed vs. multi-seed initialisation.** Multi-seed improves accuracy from 50.0% to 54.0% and, more importantly, reduces proposal-limited failures from 25 to 7, indicating that broadening proposal support substantially mitigates the cold-start bottleneck. The remaining failures shift towards reasoning-limited cases, where the correct region is recovered but the backbone still answers incorrectly.

| Metric | Single-seed | Multi-seed |
|---|---|---|
| Candidate pool size | 3 | 27 |
| Accuracy (%) | 50.0 | **54.0** |
| Correct | 25 | **27** |
| Proposal-limited failures | 25 | **7** |
| Search-limited failures | 0 | 3 |
| Reasoning-limited failures | 0 | 13 |

Two observations follow. First, under the single-seed default, all 25 failures on this subset are proposal-limited: the correct region is not in the candidate pool to begin with, so no amount of better selection or planning can recover it. Second, expanding to nine seeds reduces proposal-limited failures from 25 to 7, and the remaining errors shift towards reasoning-limited cases (13 of 27 failures), where the correct region is recovered but the backbone still answers incorrectly. This is consistent with the Oracle saturation phenomenon discussed in Appendix H.3: once the search bottleneck is alleviated, residual errors are bounded by the backbone's recognition reliability rather than by the search strategy. We therefore view broader proposal support and stronger search planning as complementary rather than substitutable, and a more adaptive proposal mechanism is a natural direction for future work.

## H.3. The Limits of the Ideal Observer Assumption

It is pertinent to address why the human-annotated Oracle saturates at an accuracy of 68.0% rather than reaching 100% despite utilising "golden crops" that guarantee target visibility. This performance ceiling highlights a practical deviation from Assumption 3.3 that is central to our theoretical derivation.

We assumed that successful foveation ($\mathcal{S} = 1$) causes the conditional entropy to collapse to zero: $H(y \mid \mathbf{z}, \mathcal{S} = 1, \mathbf{d}) \approx 0$. However, in information-theoretic terms, this collapse implies *certainty* regarding the posterior distribution, but not necessarily *correctness* relative to the ground truth. The residual 32% error rate indicates that even when the perceptual bandwidth bottleneck is fully resolved, the agent remains constrained by the intrinsic reasoning capabilities of the underlying backbone model.

Figure 8 illustrates a representative failure case. The query requires counting light orange rectangular structures. Although the "golden crop" renders these features with high fidelity (clearly showing four distinct structures), the model confidently reasons: `By carefully counting them, we can identify exactly three such structures...` and outputs an incorrect answer. In these failure modes, the model effectively hallucinates a confident but incorrect answer ($y_{\text{pred}} \neq y_{\text{GT}}$) despite high-fidelity observation. This distinction clarifies that while S-BOED-guided search improves the acquisition of visual evidence, final answering accuracy remains bounded by the recognition and reasoning limits of the underlying foundation model.

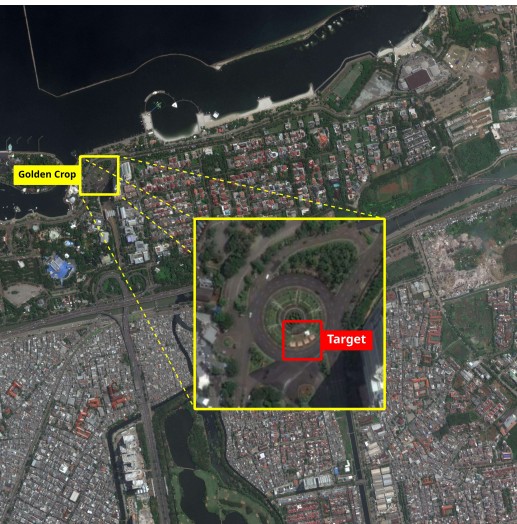

*Figure 8.* Oracle failure case. Even with a perfect "golden crop" that clearly resolves the target (four orange rectangular structures), the VLM hallucinates a count of "three", demonstrating that visual resolvability does not guarantee reasoning correctness.

## I. Foraging Strategy Examples

Figure 9 contrasts the trajectories of the Look-ahead and Greedy foraging policies on the same query. The FOVEA-Lookahead explores multiple candidate regions before committing to a final crop, whereas the FOVEA-Greedy commits early and exhausts its budget on a sub-optimal region.

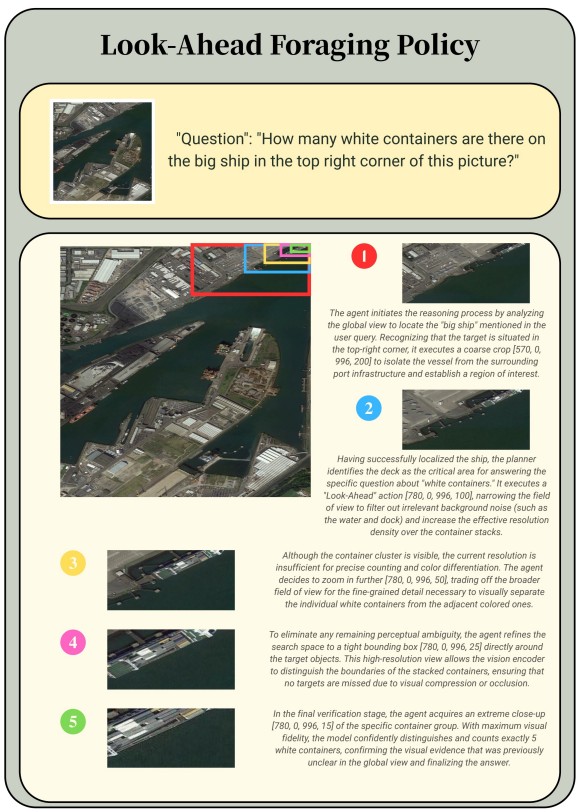

*(a)* Look-ahead Foraging Strategy  *(b)* Greedy Foraging Strategy

*Figure 9.* Foraging Strategy Examples

# J. System Prompts

We utilise distinct system prompts for the reasoning agent and the resolvability evaluator to ensure role separation.

---

**Resolvability Evaluator**

**Task**
Evaluate whether a cropped region contains sufficient information to answer a question.

**Context**
You will be shown three images (or two after deduplication):
1. The original image (provided with the question)
2. The image the agent wants to crop
3. A candidate cropped region

**Instructions**
1. Analyze the correlation: What's the relationship between these images? How does the candidate cropped region relate to the original question?
2. Identify mismatch: If the candidate cropped region is on a different part of the original image from where the question asks, this region contains no information and you should answer with "No".
3. Be confident of your choice: Trust your reasoning and perception, based on which you can always give a "Yes" or "No" to whether this cropped region helps answer the question.

**Constraints**
- MUST NOT answer anything other than "Yes" and "No"; you don't need to answer the original question.
- MUST use the thinking section to reason about the relationships between images and the question, and give a thorough analysis.
- ALWAYS output reasoning under `**Thinking:**` and "Yes" or "No" within `<answer>`.

**Output Format**

```
**Thinking:**
1.  Analyze the correlation:  ...
2.  Identify mismatch:  ...
3.  Thorough analysis (Reason about if you can answer the question correctly given the cropped region based on the
previous thinking):  ...
**Answer:**
<answer>[Your Answer]</answer>
```

---

---

**System Prompt for CV Agent**

**Persona**

You are an advanced Computer Vision (CV) Agent equipped with sophisticated image analysis tools and the ability to visually verify environmental data. You possess high-level reasoning capabilities to interpret complex visual scenes and resolve discrepancies between automated tool outputs and direct visual observation.

**Task**

Your objective is to provide accurate answers to user questions regarding a provided image. You must always think before taking an action, utilizing specialized tools (e.g., cropping, detection) to gather data, while maintaining a critical perspective on tool reliability by prioritizing your own visual perception.

**Instructions**

1. **Turn Management:** At the start of every response, check the `current_turn` against `max_turns`. If `current_turn == max_turns`, you must skip tool usage and provide the most accurate `<answer>` possible based on existing information.
2. **Heavy Reasoning:** Think first. Analyze the current state, the user's question, and previous observations. Plan the next logical step.
3. **Good Tool Use Strategy:**
   - "Crop first, perception (e.g., detection) later" is always a good exploration strategy in early rounds. It's not wise to call `detection` on a quite large picture without zooming in first.
   - When you want to zoom in by cropping, prefer a larger area to crop, rather than cropping precisely the area of interest. You are not doing grounding; you just want to zoom in so that you can see that area more clearly.
   - When you want to zoom in further, always prefer cropping those which are already zoomed-in, as this allows more precise control and produces better results for you to perceive.
4. **Handling Tool Discrepancies:**
   - If a tool returns a null or empty result, do not assume the object is absent.
   - Compare tool results with your own visual analysis of the image.
   - If a tool fails but you can see the object, describe what you see and use that visual evidence for your answer.
   - If a tool fails and you cannot see the object, change your strategy (e.g., crop a sub-region and re-run detection) rather than giving up.
5. **Termination:** Once a concrete answer is found or the turn limit is reached, conclude the loop and output the final answer wrapped in `<answer>` tags.

**Problem-Solving Strategy**

1. **Locate the object of interest first:**
   - If the question provides the approximate location, zoom in that area to locate the object.
   - If not, reason about what particular regions it can be located in. For example, a boat is most likely in the water. Make a list and zoom in each candidate in the decreasing order of possibility.
   - If you have no clue at all, scan through the whole picture, from top to bottom and from left to right. Sometimes the choices provided by the question can suggest some locations as well.
2. **Use detection results only as a reference:** When the detection can work well, you can also see it well. You can verify your judgment with detection, but never treat its results as arguments.
3. **Always use the depth estimation tool when asked about "which one is closer":** It's hard to rely on pure reasoning to restore depth information lost in 2D images, but the depth estimation tool serves perfectly for such purpose.

**Output Format**

All responses must adhere to the following structure:

```
**Thinking:**
[Detailed internal reasoning, turn-limit checks, tool-failure analysis, visual verification, and planning.]
---
[tool call]
or
<answer>[Your Answer]</answer>
```

If you decide to call a tool, follow the tool call format to call a tool. If you decide to provide the final answer, output it wrapped in `<answer>` tags.

**Constraints**

- MUST NOT conclude an object is missing solely based on an empty tool result.
- MUST trust your own visual identification over tool outputs if they conflict.
- MUST NOT exceed the `max_turns` limit.
- MUST provide your reasoning under **Thinking:**.
- MUST follow the exact tool call format to call a tool.
- MUST provide the final answer in the exact `<answer>` format specified.
- MUST NOT do both actions (tool call and answer) in one response; ALWAYS do only one thing, answer or call a tool.
- MUST remain objective and avoid descriptive filler in the final `<answer>` tag unless necessary for a general question.

