# OpenReview forum: "The Perceptual Bandwidth Bottleneck in Vision-Language Models: Active Visual Reasoning via Sequential Experimental Design"
_ICML.cc/2026/Conference — ICML 2026 regular_

### Official Review · Reviewer_xhBK · 2026-02-28

**Soundness:** 3
**Presentation:** 3
**Significance:** 2
**Originality:** 3
**Overall Recommendation:** 5
**Confidence:** 3

**Summary:**

This paper proposes an active viewing strategy for vision language models on high resolution images, where the model does not consume the full image at once but instead sequentially selects crops to examine. The method is motivated by a Bayesian experimental design view of perception under limited visual bandwidth and uses a practical proxy that balances how much of the image is covered with how clearly details can be resolved, aided by lightweight probing to score candidate crops. The approach is applied at inference time without retraining and shows consistent improvements over single pass and ReAct style baselines on several high resolution benchmarks, including remote sensing style gigapixel settings.

**Compliance With Llm Reviewing Policy:**

Affirmed.

**Final Justification:**

I think this is a solid paper.

**Key Questions For Authors:**

1. How do you choose the step budget and probing budget in practice, and do you have a principled stopping rule that adapts the number of views to question difficulty and image complexity rather than using fixed hyperparameters?

2. To what extent is performance limited by the quality and recall of the candidate crop proposal mechanism?  If the correct region is not proposed, can look-ahead planning recover at all, and how sensitive are the results to proposal diversity, number of candidates, and the proposal generator used?

3. How is information from multiple crops represented and aggregated over time to avoid redundant exploration and to maintain consistency across observations?  Do you maintain an explicit belief or state over explored regions, and how do you handle conflicting evidence from different crops when producing the final answer?

**Limitations:**

Yes

**Strengths And Weaknesses:**

**Strengths**
1. The paper reframes high-resolution VLM inference as a sequential experimental design problem, providing a coherent Bayesian perspective that motivates why iterative cropping and information-seeking actions are necessary under a fixed perceptual bandwidth.

2. The approach is model agnostic and can be applied at inference time, using a coverage and resolution trade off with lightweight probing to allocate limited views to informative regions, and it delivers consistent improvements over single pass baselines on several high resolution benchmarks.

**Weaknesses**
1. The method relies on lightweight probing questions (e.g., a Yes probability) to score candidate crops, but this signal may be poorly calibrated and can conflate genuine visual evidence with language priors or hallucinations.


2. The proxy objective assumes that including the target at sufficient resolution will translate to correct task performance, which may not hold when the backbone still misreads text, miscounts objects, or fails on relational reasoning even with clear crops. A more direct evaluation is needed to show that optimizing the proxy correlates with improvements in downstream correctness across tasks and domains.

3. A potential concern is that multi step interaction can accumulate language drift. Because the model repeatedly generates intermediate conclusions and plans for the next view, it may gradually commit to an incorrect hypothesis and then reinforce it over subsequent steps, ultimately producing a high confidence but wrong final answer.

---

> ### Author Rebuttal · Authors · 2026-03-30
>
> # Response to Reviewer xhBK
> The anonymous supplementary PDF can be accessed at: https://anonymous.4open.science/r/ICML2026-Rebuttal-21E6/icml2026_rebuttal.pdf
>
> ## **Re: Probe validity and its relation to downstream correctness**
> We do **not** treat the Yes/No probe as an exact estimator of information gain, but as a practical signal of crop utility.
>
> We performed two additional controlled analyses on 50 annotated remote-sensing position-reasoning examples. First, compared against a hard negative region and a random region, the true answer-relevant region receives much higher probe scores (**0.633 vs. 0.187 / 0.187**) and much higher answer accuracy when the model answers using the crop together with its normalised coordinates (**52.0\% vs. 10.0\% / 12.0\%**). Second, across a larger candidate pool, higher probe scores correlate with both localisation (**0.538**) and QA correctness (**0.392**), supporting that the probe is aligned with both better grounding and better downstream usefulness. The full numerical breakdown is provided in the anonymous supplementary PDF (Table 4).
>
> At the same time, we agree that sufficient resolution does **not** guarantee correctness: even with the human-annotated oracle crop, accuracy saturates at **68.0\%**, showing that backbone recognition/reasoning remains a separate bottleneck beyond search. We will clarify in the paper that the probe is intended as an empirical surrogate for **crop utility**, not as an assumption that visibility alone guarantees final correctness.
>
> ## **Re: Proposal dependence, cold-start, and less-local alternatives**
> We agree that **proposal recall is a real bottleneck** in the current single-seed local instantiation.
>
> To diagnose this directly, we ran an additional 50-example remote-sensing study comparing the original single-seed setting against a less-local multi-seed alternative. Accuracy improves from **50\% to 54\%**, while **proposal-limited** failures (i.e., the correct region never enters the candidate set) drop from **25 to 7**. This shows that the cold-start issue is real, but is primarily a limitation of the current single-seed proposal mechanism rather than of the BOED view itself. The full breakdown is provided in the anonymous supplementary PDF (Table 5).
>
> This also clarifies the role of planning: look-ahead cannot recover a region that never enters the candidate set. Proposal diversity and planning depth are therefore complementary.
>
> ## **Re: Step/probing budget and stopping rule**
> In the current implementation, both the **step budget** and the **probing budget** are fixed hyperparameters. In practice, we choose them based on the empirical accuracy-compute trade-off (see Figure 2 in the anonymous supplementary PDF): modest budgets already recover most of the gain in easier settings, while larger budgets can be useful when the task is more search-dominated.
>
> Regarding task complexity, our motivation is consistent with Remark 3.1 (“information cliff”): some questions may require sequentially complementary observations rather than a single sufficient view. We do not claim to currently have a fully adaptive online rule for detecting such cases. A more principled direction for future work is a cost-aware stopping rule, where additional views are acquired only when their expected value justifies the extra compute.
>
> ## **Re: Multi-step interaction, language drift, and aggregation across multiple crops**
> In the current implementation, we do **not** maintain an explicit belief map or structured posterior over explored regions. Instead, information from multiple crops is aggregated implicitly through the multimodal interaction history: previously viewed crops, their coordinates, and their observations are fed back into the context, and subsequent decisions are conditioned on this accumulated history.
>
> We agree that multi-step interaction can in principle accumulate drift if intermediate conclusions are wrong. However, our evidence suggests that intermediate reasoning is not merely harmful verbalisation: when we remove the model’s intermediate reasoning and provide only the visual evidence on the same 50 remote-sensing questions, performance drops substantially (**by 18 points**; see the anonymous supplementary PDF, Table 7). This indicates that current VLMs are using intermediate reasoning to integrate information across views, rather than only reinforcing earlier mistakes. We also observe qualitative cases where a one-shot prediction selects the wrong quadrant, while sequential scanning and local verification recover the correct answer (the representative example is shown in the anonymous supplementary PDF, Figure 1). Together with Appendix E, this suggests that the model is using accumulated context to revise its hypothesis over time. We will clarify in the paper that the current implementation relies on **implicit context-based aggregation**, not an explicit belief map with hand-designed conflict resolution.

---

> > ### Author Rebuttal · Reviewer_xhBK · 2026-04-01
> >
> > Thanks. I raised my socre.

---

> > > ### Author Response · Authors · 2026-04-02
> > >
> > > Thank you for your careful follow-up and for taking the time to revisit the paper after the rebuttal. We very much appreciate your thoughtful feedback.

---

### Official Review · Reviewer_MSDz · 2026-03-05

**Soundness:** 3
**Presentation:** 2
**Significance:** 2
**Originality:** 3
**Overall Recommendation:** 2
**Confidence:** 4

**Summary:**

The author derived a computationally manageable approximate solution for balancing spatial coverage and resolution, and designed an unsupervised inference strategy as the practical instance of the S-BOED objective. This strategy is compatible with any optimization algorithm such as greedy sampling and forward planning, and can be adapted to intelligent agents with multiple visual tools.

**Compliance With Llm Reviewing Policy:**

Affirmed.

**Final Justification:**

The author's rebuttal did not change my assessment. I suggest keeping the current score.

**Key Questions For Authors:**

NO

**Limitations:**

yes

**Strengths And Weaknesses:**

Advantages:

1) The active visual reasoning was formalized as an S-BOED problem, establishing a theoretical bridge between visual information search and optimal decision-making under the constraint of perceptual bandwidth. The trade-off relationship between field of view and resolution was quantified.

2) The proposed S-BOED framework is integrated into the existing VLM in the form of a plugin, without the need to retrain or fine-tune the backbone model. Moreover, the derived coverage-resolution objective function is compatible with various optimization algorithms such as greedy sampling, MCMC, and forward planning, and can be flexibly selected based on the complexity of the task.

Disadvantages:

1）The core assumption of the framework is that "when the target is successfully focused and analyzed, the semantic entropy approaches 0". However, in reality, even if high-fidelity visual features are obtained, the backbone VLM still exhibits reasoning hallucinations, resulting in the performance of the manually annotated Oracle only reaching 68.0%. This indicates that this method can only optimize the acquisition of visual evidence but cannot solve the semantic reasoning flaws of the VLM itself.

2）To approximately solve the S-BOED objective, the method employs strategies such as Monte Carlo sampling, MCMC, or forward planning, which incurs significant token costs and further increases the inference time due to the randomness of the sampling process.

3）The design of the paper has a cold-start problem. If the target is completely invisible in the global view, the VLM will assign an extremely low prior probability to the target area, and the search strategy will determine it as the background area and thus abandon the exploration.

4）Throughout the entire experiment, Qwen3-VL-30B-A3B-Instruct was used as the sole backbone VLM. No validation was conducted on VLMs with different architectures or different parameter quantities (such as small models of 7B/13B). Therefore, it is impossible to prove the adaptability of the S-BOED framework in various VLMs, nor can the correlation between model size and performance improvement of the framework be analyzed.

5）The paper did not conduct separate ablation experiments on the key components of the S-BOED framework, which prevented it from quantifying the contribution of each component to the overall performance and made it difficult to determine which components were the core factors contributing to the performance improvement.

---

> ### Author Rebuttal · Authors · 2026-03-30
>
> # Response to Reviewer MSDz
> The anonymous supplementary PDF can be accessed at: https://anonymous.4open.science/r/ICML2026-Rebuttal-21E6/icml2026_rebuttal.pdf
>
> ## **Re: Oracle performance and backbone reasoning limitations**
>
> We note that this limitation is already acknowledged in Remark 3.4, which explicitly states that the search strategy is responsible for acquiring high-fidelity evidence, while the interpretation of that evidence is delegated to the backbone VLM, and that this approximation ignores hallucination on clear images. Assumption 3.3 is therefore introduced as a tractable planning approximation, not as a claim that a correctly focused crop guarantees perfect end-to-end task performance.
>
> The Oracle result (68.0%) is consistent with this stated scope: even after the perceptual bandwidth bottleneck has been substantially alleviated by a human-annotated golden crop, the backbone still exhibits residual recognition/reasoning errors. The Oracle gap indicates that evidence acquisition and final semantic reasoning are distinct bottlenecks: S-BOED is designed to improve the former under a perceptual bandwidth bottleneck, but it does not by itself eliminate the latter.
>
>
> ## **Re: Inference cost**
> We agree that approximate search strategies such as Monte Carlo sampling, MCMC, and forward planning introduce additional token cost and latency. To make this trade-off more transparent, we provide an **accuracy-versus-compute curve** in the anonymous supplementary PDF (Figure 2). This shows that the framework is not tied to a single expensive policy: lower-cost variants provide moderate gains, while higher-cost planning can yield larger improvements in more search-dominated gigapixel settings.
>
> ## **Re: Cold-start failure mode**
> We agree that this is a real limitation of the current single-seed local instantiation. In fact, this failure mode is already analysed in our Appendix G: when the target is effectively imperceptible in the downsampled global view, the initial proposal can be badly misaligned, causing the planner to spend its finite budget on more ambiguous but incorrect regions while the true region receives too little initial support to re-enter the candidate set.
>
> To quantify this more directly, we added an additional 50-example remote-sensing diagnostic comparing the original single-seed setting against a less-local multi-seed alternative. Under the same naive local-refinement budget, accuracy improves from **50\% to 54\%**, while **proposal-limited** failures drop from **25 to 7**, where “proposal-limited” means that the correct region never enters the candidate set in the first place. This suggests that the cold-start issue is real, but is primarily a limitation of the current single-seed proposal mechanism rather than of the S-BOED view itself. Broader proposal support can substantially mitigate this bottleneck. The full breakdown is provided in the anonymous supplementary PDF (Table 5).
>
> ## **Re: Validation on only one backbone VLM**
> To address it directly, we additionally evaluated the framework on a substantially smaller open VLM, **Qwen3-8B-VL**. The same overall trend remains across all five benchmarks: our method is competitive on CVBench and outperforms both Direct and ReAct on the other four datasets, improving the mean score from **70.9 / 72.5** (Direct / ReAct) to **74.9** (Ours). This suggests that the framework is not specific to a single 30B backbone. We agree that this is not yet a full cross-architecture study, nor is it sufficient to support a strong claim about model-size scaling trends. The full benchmark-wise breakdown is provided in the anonymous supplementary PDF (Table 1).
>
> ## **Re: Lack of separate ablation experiments on key components**
>
> We agree that clearer component-level attribution would strengthen the paper. Since S-BOED is an inference-time decision framework rather than a monolithic trainable architecture, standard module-removal ablations are less natural here; instead, we added targeted component-level analyses on the key components (Tables 4-7). In the original paper, the most direct ablations are the comparisons across search strategies (Naive / MCMC / Look-ahead), which isolate the effect of increasingly stronger planning under the same framework.
>
> To further clarify the probe component, we added targeted diagnostics rather than a full factorial ablation: oracle regions receive much higher probe scores than distractor/random regions (**0.633 vs. 0.187 / 0.187**) and much higher downstream QA accuracy (**52.0\% vs. 10.0\% / 12.0\%**), while a controlled selector ablation on a fixed 3$\times$3 candidate pool shows that probe-based selection outperforms both direct VLM selection and random selection in QA accuracy (**30\% vs. 20\% / 10\%**) and localisation (**52\% vs. 34\% / 12\%**). These additions are reported in the anonymous supplementary PDF (Tables 4-7).

---

> > ### Author Rebuttal · Reviewer_MSDz · 2026-04-04
> >
> > Thank you for the author's reply. I suggest maintaining the current score.

---

### Official Review · Reviewer_qHSd · 2026-03-13

**Soundness:** 3
**Presentation:** 3
**Significance:** 3
**Originality:** 3
**Overall Recommendation:** 4
**Confidence:** 4

**Summary:**

This paper studies active visual reasoning for high-resolution vision-language models under a perceptual bandwidth bottleneck. The paper formulates sequential crop selection as a sequential Bayesian optimal experimental design problem, derives a tractable coverage-resolution objective as a proxy for expected information gain, and instantiates the framework as a training-free inference pipeline with local probing, greedy search, and look-ahead planning.

**Compliance With Llm Reviewing Policy:**

Affirmed.

**Final Justification:**

The additional clarifications and experimental results has fully addressed my concerns. Thus I keep my score as "weak accept".

**Key Questions For Authors:**

The formal derivation centers on explicit beliefs, expected information gain, and the coverage-resolution objective. In the implemented system, what exactly should be interpreted as the approximation to the spatial belief, the resolution probability, and the coverage term? Can the authors provide stronger evidence that the “Yes/No” probe is a faithful estimator of downstream information gain rather than just answerability confidence?

The practical search procedure appears strongly tied to the initial crop proposal. Could the authors quantify how often failures are due to poor initialization, and compare against a less local alternative that does not rely on a seed proposal? This seems especially important given the paper’s own cold-start failure mode.

Could the authors provide more direct comparisons to closely related high-resolution / active-perception methods, especially SEAL/V* and RAP, or explain why such comparisons are not feasible? Evaluation on a dedicated active-perception benchmark such as ActiView would also strengthen the paper substantially.

The main benchmark gain over the same-backbone ReAct baseline is relatively modest, while the stronger look-ahead gains come with large token overhead. Could the authors provide accuracy-versus-compute curves, latency measurements, or normalized efficiency metrics so readers can better assess the practical tradeoff?

The framework is presented as flexible and plug-and-play, but the experiments use only one backbone and one tool stack. Does the method transfer to another open VLM, and how sensitive is it to the specific resolvability prompt, tool set, or Monte Carlo sampling parameters?

**Strengths And Weaknesses:**

Postive:
This paper tackles a meaningful and timely problem. High-resolution perception remains a real weakness of current VLM agents, and the paper’s framing around perceptual bandwidth is intuitive and well motivated. I also appreciate that the submission does more than present a heuristic prompt recipe: it attempts to provide a decision-theoretic lens, includes a focused oracle analysis to separate search failures from recognition failures, and provides qualitative failure analysis for the look-ahead planner.

Weaknesses:
My main concern is the gap between the formalism and the actual algorithmic realization. The theory is built around explicit beliefs, expected information gain, a coverage-resolution utility, and Bayesian posterior updates. In practice, however, the implementation replaces these quantities with implicit in-context belief tracking and a binary resolvability probe, where the utility is approximated by the model’s probability of answering “Yes” for a candidate crop. This jump is supported by several strong assumptions, including factorized spatial-semantic beliefs, calibrated visibility, and an ideal-observer entropy-collapse approximation. As a result, the method feels much closer to a well-motivated local crop-refinement heuristic than to a faithful instantiation of sequential Bayesian experimental design

the empirical cost-benefit tradeoff. On the full benchmark suite, the same-backbone gain over ReAct is real but modest, improving the mean score from 75.1 to 77.7. The more substantial improvement appears on the remote-sensing subset, where the look-ahead planner reaches 54.7 versus 45.1 for ReAct, but this comes with a very large inference overhead: roughly 9.5× input tokens and 55.2× output tokens in the reported cost table. This does not invalidate the method, but it makes the practical significance less clear than the paper’s presentation suggests.

---

> ### Author Rebuttal · Authors · 2026-03-30
>
> # Response to Reviewer qHSd
>
> The anonymous supplementary PDF can be accessed at: https://anonymous.4open.science/r/ICML2026-Rebuttal-21E6/icml2026_rebuttal.pdf
>
> ## **Re: Formalism vs. implementation**
>
> We agree that the current system is **not** an exact sequential BOED solver with an explicit posterior map and exact EIG computation. Rather, it should be understood as a **practical surrogate instantiation** of the S-BOED view. Concretely, the formal spatial belief $p_t(\ell)$ is represented **implicitly** through the history-conditioned multimodal context $H_t$, rather than an explicit belief grid (Sec. 4.1 and Appendix E). In implementation, we do not estimate coverage and resolution as two separate quantities; instead, the crop design $d=[u,v,w,h]$ jointly determines both field of view and effective resolution, and the Yes/No probe serves as a single empirical surrogate for the **combined coverage-resolution objective** in Eq. (4). We will revise the wording to make this surrogate status explicit.
>
> ## **Re: Stronger evidence for the Yes/No probe**
>
> We do **not** claim that the Yes/No probe is an exact estimator of information gain. Instead, we use it as a practical signal of crop utility. To test whether it is more than generic answerability confidence, we ran additional controlled analyses showing that higher probe scores are associated with both more answer-relevant crops and better downstream answering. Since Reviewer **xhBK** raised the same concern, we report the full numerical evidence there and provide the full breakdown in the supplementary PDF (Table 4). We will clarify that the probe is an empirical surrogate for crop utility rather than an exact estimator of information gain.
>
> ## **Re: Poor initialisation / less-local alternatives**
>
> We agree that poor initialisation is a real bottleneck in the current single-seed local search. To quantify this directly, we ran an additional 50-example remote-sensing diagnostic comparing the original single-seed proposal against a less-local multi-seed alternative; the detailed breakdown is given in the supplementary PDF (Table 5). Briefly, accuracy improves from **50% to 54%**, while **proposal-limited** failures drop from **25 to 7**, indicating that the correct region often fails to enter the candidate pool under single-seed initialisation and that broader proposal support can materially mitigate this cold-start failure mode.
>
> ## **Re: RAP/SEAL/V\* and ActiView**
>
> We additionally evaluated against **RAP** and on the dedicated active-perception benchmark **ActiView**. The main trend remains consistent: on ActiView, adding S-BOED-style micro-crop refinement improves over the vanilla pipeline, with **Look-ahead reaching 63.69\% vs. 62.46\% for Vanilla** and **MCMC reaching 63.08\%**. We also compare to **RAP**; while RAP is competitive on some benchmarks, the overall mean on our suite still favours our method (**77.7** for Ours vs. **71.9** for RAP), with the full per-benchmark breakdown in the anonymous supplementary PDF (Table 2 and 3).
>
> We agree that SEAL / V* is highly relevant prior work, but it is not a strictly comparable baseline under our setting. Our method isolates the crop-planning component under a fixed backbone and tool interface, whereas SEAL / V* changes the overall system set-up more broadly. It is therefore not a drop-in comparison under the same protocol.
>
> ## **Re: Accuracy-versus-compute trade-off**
>
> To make the practical trade-off more transparent, we provide an **accuracy-versus-compute curve** on an extra 50-example remote-sensing subset, where search is the dominant bottleneck. The trend is monotonic: as search budget increases, accuracy also increases. The full curve is shown in Figure 2 of the supplementary PDF. While this is not a full latency study, it directly shows the compute-accuracy frontier of stronger search policies in this high-resolution regime.
>
> ## **Re: Transfer and sensitivity**
>
> To address transfer directly, we additionally evaluated the method on a substantially smaller open VLM, **Qwen3-8B-VL**, and observed the same overall trend across all five benchmarks: our method remains competitive on CVBench and outperforms both Direct and ReAct on the other four datasets, improving the mean score from **70.9 / 72.5** (Direct / ReAct) to **74.9** (Ours). The full benchmark-wise breakdown is provided in the supplementary PDF (Table 1). This suggests that the method is not specific to a single large backbone.
>
> For sensitivity, our additional accuracy-versus-budget study shows a clear compute-accuracy tradeoff for search-budget / Monte-Carlo-style candidate generation. For the specific resolvability prompt and tool stack, we agree that current evidence is still limited, and we will clarify this limitation more explicitly in the paper.

---

> > ### Author Rebuttal · Reviewer_qHSd · 2026-04-06
> >
> > Thank you to the authors for the additional clarifications.

---

### Decision · Program_Chairs · 2026-04-30

**Decision:**

Accept (regular)

**Comment:**

In general, the reviewers like the concept of perceptual bandwidth and the way the paper formulates the high-resolution perception problem and instantiates the framework using VLM. In the meantime, the reviewers cite concerns about the gap between the theory and implementation, high inference cost compared with the ReAct baseline, cross-backbone evaluation, and others. The rebuttal addresses most of these concerns, leading to 2x (weak) accept and 1x reject ratings. The lingering concerns discussed extensively among the reviewers are 1) whether the ideal observer assumption (Assumption 3.3) is realistic, 2) lack of evidence for implicit Bayesian update, and 3) high inference cost. One reviewer mentions that (1) and (2) are fairly discussed in Appendix G.1 and E respectively. The AC checks the paper and believes that the discussions provided in the Appendix are sufficient and helpful for researchers and practitioners to build on top of this research. Regarding the inference cost, Table 5 indeed shows that the proposed approach requires more tokens for thinking and tool calls but the accuracy gains should not be ignored—there is a trade-off between token usage and accuracy. The high inference cost could be viewed as a limitation of the approach but not grounded for rejecting this work, which provides a new agentic framework for solving the perception problem in VLMs. The AC recommends that the paper be accepted given that this work has merits.